# Ribosomal profiling during prion disease uncovers progressive translational derangement in glia but not in neurons

**Claudia Scheckel\*, Marigona Imeri, Petra Schwarz, Adriano Aguzzi\***

Institute of Neuropathology, University of Zurich, Zurich, Switzerland

**Abstract** Prion diseases are caused by PrP^Sc, a self-replicating pathologically misfolded protein that exerts toxicity predominantly in the brain. The administration of PrP^Sc causes a robust, reproducible and specific disease manifestation. Here, we have applied a combination of translating ribosome affinity purification and ribosome profiling to identify biologically relevant prion-induced changes during disease progression in a cell-type-specific and genome-wide manner. Terminally diseased mice with severe neurological symptoms showed extensive alterations in astrocytes and microglia. Surprisingly, we detected only minor changes in the translational profiles of neurons. Prion-induced alterations in glia overlapped with those identified in other neurodegenerative diseases, suggesting that similar events occur in a broad spectrum of pathologies. Our results suggest that aberrant translation within glia may suffice to cause severe neurological symptoms and may even be the primary driver of prion disease.

**\*For correspondence:**
claudia.scheckel@usz.ch (CS);
adriano.aguzzi@usz.ch (AA)

**Competing interests:** The authors declare that no competing interests exist.

## Introduction

Prion diseases (PrD) are fatal neurodegenerative diseases that are caused by transmissible proteinaceous particles termed prions (*Scheckel and Aguzzi, 2018*). Prions consist primarily of PrP^Sc, pathological aggregates of the cellular prion protein (PrP^C) and can be highly infective. PrDs are conventionally classified as acquired, genetic and sporadic forms of the disease. While the cause of PrP^C aggregation differs, all PrDs show a similar disease progression characterized by a pronounced vacuolation phenotype, activation of microglia and astrocytes, and ultimately neuronal loss. Prions are most efficiently transmitted within one and the same animal species, yet PrDs have been described in numerous mammals and are remarkably similar. In particular, mouse models of PrD appear to recapitulate most aspects of the human disease including transcriptomic changes – a finding that sets them apart from other murine models of neurodegenerative disease (*Burns et al., 2015*). Importantly, PrDs share many hallmarks with other more common neurodegenerative diseases including Alzheimer's and Parkinson's disease, and genes linked to neurodegeneration in humans have been shown to be conspicuously downregulated in PrD mice (*Burns et al., 2015*). This suggests that lessons learned from the PrD mouse model may be relevant to our understanding as well as to the development of diagnostics and therapeutics in other neurodegenerative diseases.

Several techniques have aimed at characterizing the molecular changes underlying neurodegeneration. One major hurdle when studying pathologies of the central nervous system (CNS) is selective vulnerability: the brain consists of many different cell types, yet most diseases only affect a subset of cells. The isolation of specific cell types followed by bulk sequencing (*Zamanian et al., 2012*) and single-cell sequencing (*Keren-Shaul et al., 2017*) is often used to characterize CNS diseases and injuries, and has identified disease-associated signatures in different cell types. Yet, single-cell sequencing only allows for the assessment of highly-expressed genes. Moreover, all of these approaches require prolonged dissociation and cell sorting protocols, which can introduce artefacts. Alternatively, cell-type-specific profiles can be generated from mice expressing tagged ribosomes in

the cell type of interest via translating ribosome affinity purification (TRAP) followed by RNA sequencing (*Heiman et al., 2014*). RiboTag (RT) mice expressing HA tagged ribosomes in a Cre recombinase-dependent manner (*Sanz et al., 2009*), have been used to profile multiple CNS cell populations without tissue manipulation procedures (*Boisvert et al., 2018*; *Furlanis et al., 2019*; *Haimon et al., 2018*). The datasets generated from these experiments are likely to reflect the endogenous expression of mRNA very closely.

Here, we have characterized cell-type-specific molecular changes in PrD using a combination of TRAP and ribosome profiling. The select expression and isolation of tagged ribosomes, followed by the generation and sequencing of ribosome-protected fragments (RPFs) without any preanalytical manipulations enables the assessment of endogenous translation in a quantitative, genome-wide and cell-type-specific manner. Profiling of control and PrD mice throughout disease progression revealed vast prion-induced translational changes in microglia and astrocytes, but surprisingly few changes in neurons. This suggests that the prion-induced molecular phenotypes reflect major glia alterations, whereas the behavioral phenotypes may be ascribed to just few neuronal changes or changes that were undetectable in our assay such as altered neuronal connectivity.

## Results

### Generation of mice that express GFP-tagged ribosomes in PrD-relevant cells

To characterize cell-type-specific molecular changes during PrD progression in a genome-wide manner, we used a mouse strain that expresses a GFP-tagged version of a large ribosomal protein, RPL10a, in a Cre-dependent manner (*Zhou et al., 2013*). We bred this strain with several 'Cre driver lines', expressing Cre recombinase under the control of the *Camk2a*, *Pvalb*, *Gfap* or *Cx3cr1* promoters (*Fuchs et al., 2007*; *Gregorian et al., 2009*; *Tsien et al., 1996*; *Yona et al., 2013*), to induce the expression of GFP-tagged ribosomes in excitatory CamKIIa neurons, inhibitory parvalbumin (PV) neurons, astrocytes or microglia, respectively (*Figure 1a*). Double-transgene-positive mice were intraperitoneally injected with RML6 prions or control brain homogenates and sacrificed at the indicated time points during disease progression (*Figure 1a* and *Supplementary file 1*). Brains or brain regions were dissected and immediately flash frozen in liquid nitrogen for cell-type-specific ribosome profiling, or formalin-fixed and paraffin-embedded for immunohistochemical analyses.

To confirm a cell-type-specific expression of GFP-tagged ribosomes, we examined the brains of mice by immunofluorescence with anti-GFP antibodies. Neuronal CamKIIa and PV Cre drivers have previously been shown to specifically label the corresponding neuronal subpopulation. Consistently, the CamKIIa-driven GFP expression was detectable in all CamKIIa-positive neurons in hippocampus and cortex (*Figure 1b* and *Figure 1—figure supplement 1a*) and PV-driven GFP expression was seen in all PV$^+$ cells, most prominently in the cerebellum (*Figure 1c* and *Figure 1—figure supplement 1b–c*). Co-staining with the neuronal marker NeuN, which is expressed in most neurons, confirmed that the CamKIIa promoter induced GFP expression in only a subset of NeuN$^+$ neurons. In contrast, GFP-expressing PV neurons were typically NeuN negative. These results show that neuronal GFP expression was limited to the cell type of interest in different brain regions. Importantly, the expression of GFP was strictly Cre-dependent (*Figure 1b–c* and *Figure 1—figure supplement 1*) indicating that the GFP:Rpl10a mouse strain does not exhibit leaky constitutive GFP expression.

Glia-specific Cre drivers have proven more difficult to generate, and many of them show extensive neuronal expression. The mGFAP line 77.6 utilized in this study performs better than other astrocytic Cre drivers and expresses Cre almost exclusively in astrocytes (*Gregorian et al., 2009*). Indeed, we predominantly observed GFP expression in astrocytes (*Figure 1d* and *Figure 1—figure supplement 1d*) and, to a lower extent, in neurons (arrows in *Figure 1d*). Also, the microglia Cre driver line, Cx3cr1$^{Cre}$ has been shown to display Cre-induced rearrangements not only in microglia but also in neurons, most likely due to Cx3cr1 promoter activity during neuronal development (*Haimon et al., 2018*). We therefore utilized an inducible Cx3cr1$^{CreER}$ line and activated Cre expression in microglia by treating adult mice for four weeks with tamoxifen-containing chow. While these mice showed GFP expression in microglia and not in neurons (*Figure 1e* and *Figure 1—figure supplement 1e*), they have been reported to additionally express Cre in non-parenchymal macrophages (*Goldmann et al., 2016*), corresponding to roughly ~1% of Cre-positive cells in Cx3cr1$^{CreER}$ mice

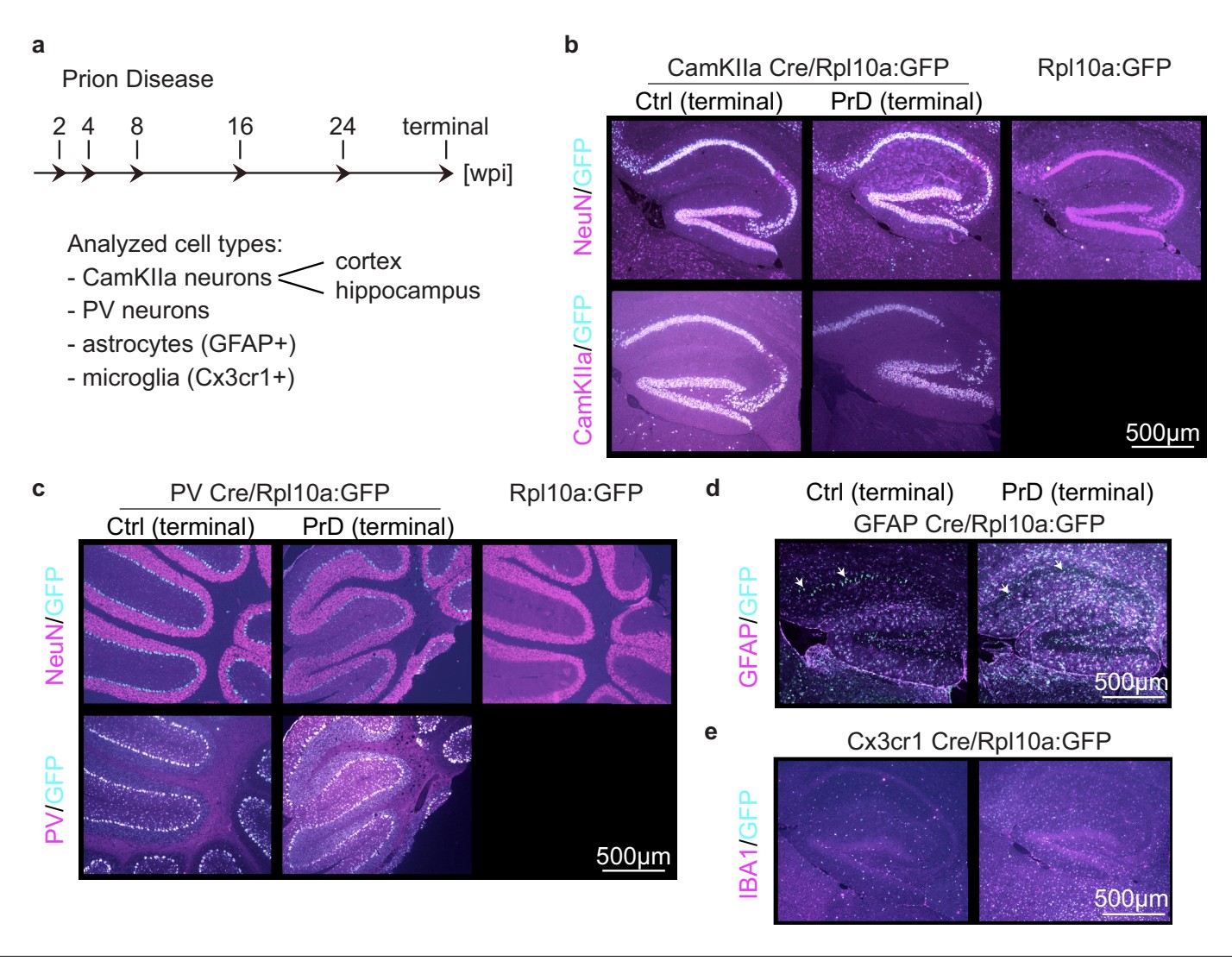

**Figure 1.** Generation of mice that express GFP-tagged ribosomes in prion disease relevant cells. (**a**) Schematic displaying which cell types were analyzed at which time point during prion disease (PrD) progression. (**b**) Brain sections of control (ctrl) and terminally PrD CamKIIa/Rpl10a:GFP mice were stained for GFP and CamKIIa or NeuN. A subset of NeuN+ cells and all CamKIIa+ cells showed Rpl10a:GFP expression in the hippocampus. No GFP signal was detectable in Rpl10a:GFP mice (not bred with a Cre expressing strain). (**c**) Brain sections of ctrl and PrD PV/Rpl10a:GFP mice were stained for GFP and PV or NeuN. Cells that were NeuN-negative and PV+ showed Rpl10a:GFP expression in the cerebellum. No GFP signal was detectable in Rpl10a:GFP mice (not bred with a Cre expressing strain). (**d**) Brain sections of ctrl and PrD mice expressing Rpl10a:GFP in GFAP+ cells were stained for GFP and GFAP. (**e**) Brain sections of terminally diseased and control mice expressing Rpl10a:GFP in Cx3cr1+ cells were stained for GFP and IBA1.

The online version of this article includes the following figure supplement(s) for figure 1:

**Figure supplement 1.** Generation of mice that express GFP-tagged ribosomes in prion disease relevant cells.

(*Haimon et al., 2018*). The expression of GFP in non-glia cell types, was taken into consideration in subsequent analyses.

We further assessed expression changes of GFP and different cell markers between control and prion-injected brains at the terminal stage of PrD. Hallmarks of PrD include neuronal loss as well as pronounced micro- and astrogliosis. Consistently, glial GFP, IBA-1 (microglia marker) and GFAP (astrocyte marker) levels were markedly increased in terminal PrD brains. In contrast, we did not observe a clear difference in neuronal GFP, NeuN, CamKIIa or PV expression (*Figure 1* and

*Figure 1—figure supplement 1*). This is consistent with previous observations from our lab, where no or only a minor decrease in neuronal cell number was evident at the terminal PrD stage in mice.

## Analysis of different CNS cells via cell-type-specific ribosome profiling

Typically, mice expressing tagged ribosomes are analyzed via immunoprecipitation and RNA sequencing to identify which transcripts are translated in a given cell (*Heiman et al., 2014*). In contrast, ribosome profiling yields an additional layer of information by determining the number of ribosome-protected fragments (RPFs). The number of RPFs is linked to the number of ribosomes per transcript and therefore to the translation rate. The combination of tagged ribosome immunoprecipitation and ribosome profiling thus allows the quantitative assessment of mRNA translation specifically in the cell types of interest. We therefore investigated mice expressing GFP-tagged ribosomes in different PrD-relevant cells at multiple disease stages via cell-type-specific ribosome profiling (all analyzed samples are detailed in *Supplementary file 1*). Briefly, RPFs were isolated via partial RNase digestion and GFP immunoprecipitation and subjected to library preparation and high-throughput sequencing (*Figure 2a*). Unique molecular identifiers (UMIs) were attached to each cDNA molecule to eliminate PCR duplicates, and only unique reads were included in all downstream analyses. To analyze PV neurons, microglia and astrocytes, we utilized one hemisphere per library. Pooled cortices and hippocampi of one mouse were assessed to examine CamKIIa neurons. We additionally profiled cortex, hippocampus and hemispheres of Cre-negative mice, which did not express any GFP. We typically obtained 5–10 million unique reads per library (*Figure 2—figure supplement 1a* and *Supplementary file 1*), of which only a minority ($\leq$10%) mapped to ribosomal RNA (*Figure 2—figure supplement 1b* and *Supplementary file 1*). Most aligned reads were coding sequence (CDS) reads (*Figure 2b* and *Supplementary file 1*), which is consistent with the ribosome occupancy of transcripts. Importantly, despite the increase in glia cells at the terminal stage, the libraries of all terminally diseased mice (green data points in *Figure 2* and *Figure 2—figure supplement 1*) were non-distinguishable from other libraries. One single library did not pass our quality criteria ($>10^6$ unique reads and >70% of CDS reads) and was excluded from downstream analyses (encircled in pink in *Figure 2* and *Figure 2—figure supplement 1*). While representative Cre$^+$ libraries showed a strong increase in RPF coverage at the translational start codon that tailed off, the excluded library showed a pattern that was more similar to Cre-negative libraries (*Figure 2—figure supplement 1c*).

We proceeded with summarized RPF counts per gene, referred to as RNA translation throughout the manuscript. Focusing initially on libraries generated from control mice at the earliest time point (~3 months old mice), we visualized the relationship between the different libraries. A principal-component analysis (PCA) and hierarchical clustering based on Euclidean distances revealed a segregation of the different cell types as well as a similarity between the analyzed neuronal cell types (*Figure 2c–d* and *Supplementary file 2*). Cre-negative samples most likely recapitulate the average translation in the analyzed region. With astrocytes being a highly abundant cell type in the CNS (*Miller, 2018*), it was not surprising that Cre-negative samples were most similar to GFAP samples (*Figure 2c*). Cre-negative samples were nonetheless clearly distinguishable from GFAP and other Cre$^+$ samples (*Figure 2d*).

## Identification of preferentially translated transcripts in different cell types

To confirm that we profiled the desired cell population, we compared our data with previously published studies. We defined the top 100 enriched genes in neurons (N), endothelial cells (EC), oligodendrocytes (OL), astrocytes (AS) and microglia (MG) based on RNA sequencing data generated from these cell populations (*Zhang et al., 2014*). These cell types were purified from P7 mouse brains via cell dissociation, immunopanning and FACS, and were analyzed via RNAseq. In contrast, our datasets were generated via ribosome profiling of flash-frozen brain regions from 13-week-old mice. Nonetheless, we observed that the top 100 enriched genes showed the highest translation in the corresponding RP dataset (*Figure 3a* and *Supplementary file 3*). We additionally assessed cell-type-specific markers, defined by single-cell sequencing of the visual cortex from 2-month-old mice (*Hrvatin et al., 2018*). As expected, marker genes for excitatory neurons, PV neurons, astrocytes and microglia showed the highest translation in the respective cell type (*Figure 3—figure supplement 1* and *Supplementary file 3*). The comparison further revealed that we predominantly profiled

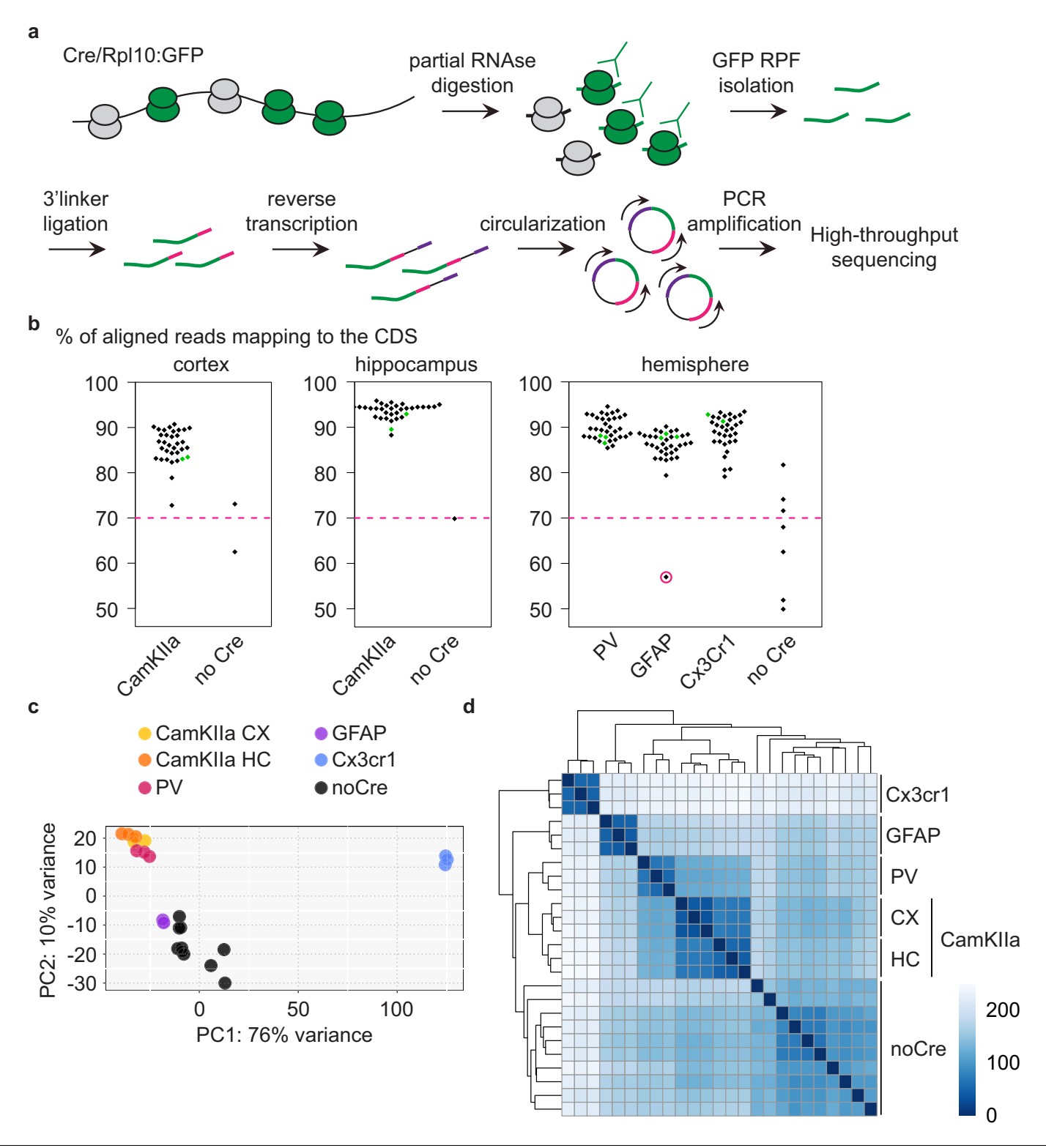

**Figure 2.** Analysis of different CNS cells via cell-type-specific ribosome profiling. (**a**) Schematic depicting the workflow of cell-type-specific ribosome profiling. After partial RNAse digestion, GFP ribosome protected fragments (RPF) were isolated via GFP immunoprecipitation and ultracentrifugation, and subjected to 3' linker ligation, reverse transcription, circularization and PCR amplification before being submitted for high-throughput sequencing. (**b**) Beeswarm plots displaying the % of reads mapping to the coding sequence (CDS). Samples are grouped according to analyzed regions and Cre drivers. Only Cre⁺ samples with a minimum of 70% CDS reads (indicated by a dashed pink line) were analyzed. One Cre⁺ sample did not pass

*Figure 2 continued on next page*

*Figure 2 continued*

quality control and was therefore excluded (encircled in pink). Terminal PrD samples are colored in green. (**c**) Principal component analysis of noCre samples and control samples at 2 weeks post-inoculation (wpi) based on rlog transformed RPF counts per gene revealing a separation according to Cre driver (~cell type), a resemblance of GFAP samples and noCre samples and similarities between neuronal cell types. (**d**) Hierarchical clustering of noCre samples and control samples at two wpi based on Euclidean distances. Heatmaps depicting the sample distance at each time point based on rlog transformed RPF counts per gene. Samples cluster according to Cre driver (~cell type) and noCre samples are clearly distinguishable.

The online version of this article includes the following figure supplement(s) for figure 2:

**Figure supplement 1.** Analysis of different CNS cells via cell-type-specific ribosome profiling.

subsets of excitatory neurons (L23, L5_1 and L5_3) and microglia (Micro_1). Additionally, macrophage markers were highly translated in the Cx3cr1 RP dataset, confirming previous observations that a minor fraction of Cx3cr1$^{CreER}$-positive cells are non-parenchymal macrophages. Collectively, these results indicate that we indeed analyzed the desired cell population.

We next identified genes that were differentially translated between the different cell types using DESeq2 (*Figure 3—figure supplement 2* and *Supplementary file 2*). We defined preferentially translated genes (PTGs) in a specific cell type if they showed an absolute log2 fold change |log2FC| of more than two and an FDR of less than 0.05 in one cell type compared to the four other cell types. This yielded a total of 2692 PTGs, including 57 CamKIIa_CX, 50 CamKIIa_HC, 299 PV, 963 GFAP and 1323 Cx3Cr1 PTGs (*Figure 3b*). A heatmap displaying the translation of the 2692 PTGs shows a preferential translation of these genes in the respective cell type and that samples clustered according to cell type (*Figure 3c*).

Several studies have previously assessed ribosome-associated RNAs in the same cell types (*Boisvert et al., 2018*; *Furlanis et al., 2019*; *Haimon et al., 2018*). Using the RiboTag (RT) mouse, which expresses HA-tagged ribosomes in a Cre-dependent manner the authors sequenced the immunoprecipitated RNA directly. In contrast, the combination of ribosome immunoprecipitation and ribosome profiling (RP) allowed us to assess cell-type specific translation in a quantitative manner. When comparing all RT and RP datasets with each other (*Supplementary file 3*) we observed the highest correlation between datasets that were generated from the same cell type (*Figure 3d* and colored comparisons in *Figure 3—figure supplement 3a*), and PTGs were most highly expressed in the corresponding cell type (*Figure 3—figure supplement 3b*). While RT and RP datasets of each cell type were overall highly correlated (*Figure 3d*), we also observed that a number of genes deviated from the linear regression line, including several PTGs (colored in *Figure 3d*). On one hand this might reflect differences linked to the analyzed tissue or the library preparation, yet some are likely to correspond to genes that are characterized by a high or low translation rate. The deviation of RT and RP datasets was particularly evident for PV and Cx3cr1$^+$ cells, which for PV can be at least partially ascribed to the analysis of different tissues (neocortex for RT vs a whole hemisphere for RP). In contrast, the same tissue (whole hemispheres) was analyzed to assess Cx3cr1$^+$ cells via RT and RP, and, our RP dataset showed a higher correlation with an RNAseq dataset generated from sorted microglia than with the RT dataset from the same study. This indicates that the GFP immunoprecipitation followed by ribosome profiling might yield more specific results compared to HA immunoprecipitation followed by direct sequencing.

## Cell-type-specific changes become only evident at late PrD stages

To discover PrD-related molecular changes, we identified differentially translated genes (DTGs: |log2FC| > 1 and FDR < 0.05) in the previously mentioned cell types at different stages during disease progression (*Figure 1a* and *Supplementary files 4–8*). Prion-induced changes only became evident at later disease time points, and except for few individual genes we could not detect any DTGs in any of the cell types at 2, 4, 8, and 16 weeks post prion inoculation (wpi). A PCA revealed a clear separation of control and PrD samples only at the terminal time point, most strikingly in GFAP and Cx3cr1$^+$ cells (*Figure 4—figure supplement 1*). Consistently, we observed most prion-induced changes in GFAP$^+$ and Cx3cr1$^+$ cells at the terminal stage (*Figure 4a* and *Figure 4—figure supplement 2*; all significantly changing genes are colored in *Figure 4—figure supplement 2*; significantly changing genes with |log2FC| > 1 are summarized in *Figure 4a*). Both of these cell types already showed several translational changes at the pre-terminal stage, 24 wpi. Upon comparing prion-

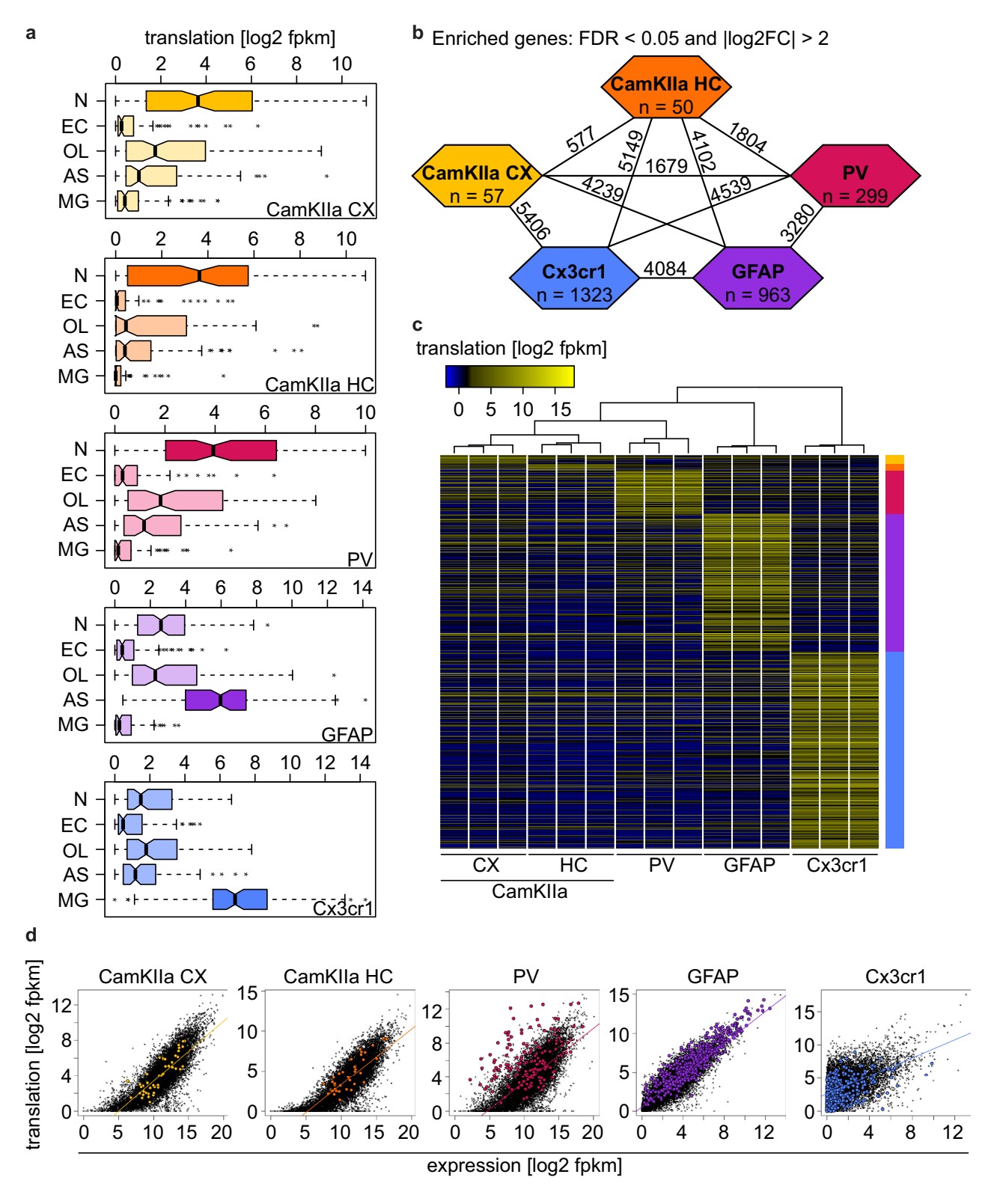

**Figure 3.** Identification of preferentially translated transcripts in different cell types. (a) Boxplots displaying the translation (log2 transformed fpkm values) of the top 100 genes in neurons (N), endothelial cells (EC), oligodendrocytes (OL), astrocytes (AS) and microglia (MG) in the indicated cell type. Boxes containing genes expected to be highly translated are shaded in a darker tone. (b) Identification of genes that were differentially translated (| log2FC| > 2 and FDR < 0.05) between two different cell types (numbers on lines) and of preferentially translated genes (PTGs) that were differentially

*Figure 3 continued on next page*

Figure 3 continued

translated in one versus all other cell types (numbers in colored hexagons). (c) Heatmap displaying the translation (log2 transformed fpkm values) of 2692 PTGs, revealing a clustering of samples according to cell type. (d) Scatterplots comparing the translation (log2 transformed fpkm values) and the expression (assessed via ribosome immunoprecipitation followed by sequencing; log2 transformed fpkm values) in the indicated cell type. Translation and expression of CamKIIa- and Cx3cr1-positive cells were assessed in corresponding brain regions. In contrast, PV and GFAP translation was examined in the whole brain, whereas PV expression was analyzed in neocortex and GFAP expression corresponds to the averaged expression across multiple regions (motor/somatosensory/visual cortex, cerebellum and hypothalamus). Linear regression lines are indicated and PTGs in the respective cell types are colored.

The online version of this article includes the following figure supplement(s) for figure 3:

**Figure supplement 1.** Marker genes are highly translated in the corresponding cell type.

**Figure supplement 2.** Identification of differentially translated transcripts in different cell types.

**Figure supplement 3.** Comparison of cell-type-specific expression and translation.

induced changes at the last two time points, we observed a correlation in both GFAP$^+$ and Cx3cr1$^+$ cells and a high overlap between the changes at the two time points, especially among upregulated genes (*Figure 4b–c*). This indicates that the pronounced translational changes in astrocytes and microglia become evident 2 months before the terminal stage, long before any PrD-related symptoms manifest. Both astrocyte activation and microglia proliferation are hallmarks of PrD, yet due to the nature of our assay, these changes reflect translational changes within the respective cell type and are not due to any differences in cell numbers.

Both glia cell types have been linked to numerous diseases and display characteristic transcriptional signatures in response to disease or other stimuli. Reactive A1 astrocytes are induced upon lipopolysaccharide treatment, microglia activation and ischemia, are characterized by the upregulation of complement cascade genes and are thought to contribute to neuronal death (*Liddelow et al., 2017*; *Zamanian et al., 2012*). Not surprisingly, we found A1 signature genes to increase with disease progression (*Figure 4d* and *Supplementary file 7*). We also observed an increase in A2 signature genes, corresponding to neurotrophic factors (*Liddelow et al., 2017*; *Zamanian et al., 2012*). The categorization of microglia into homeostatic and disease-associated (as well as other) microglia has largely replaced the initial categorization into pro-inflammatory M1 and anti-inflammatory M2 microglia (*Dubbelaar et al., 2018*). We have therefore opted for the more current categorization. Consistent with our observations for A1 genes, disease-associated microglia (DAM) genes increased during disease progression. In contrast, homeostatic microglia markers started to show a slight decrease at 24 wpi, that was more pronounced at the terminal stage of the disease (*Figure 4e* and *Supplementary file 8*). Similar observations have been reported for multiple neurodegenerative diseases (*Keren-Shaul et al., 2017*), confirming that the appearance of DAMs and the concomitant loss of homeostatic microglia are hallmarks of multiple neurodegenerative diseases.

Surprisingly, neuronal cell types displayed very few translational changes even at the terminal stage (*Figure 4a*), and we observed only a low correlation of prion-induced changes at the last two time points (*Figure 4—figure supplement 3*). This is particularly interesting as PrD is a neurodegenerative disease. As previously mentioned, our protocol does not allow us to assess the total number of analyzed cells, suggesting that the CamKIIa and PV neurons which are present in terminally diseased mice, show only minor translational changes, especially compared to astrocytes and microglia. Several of the genes that changed in neurons at 24 wpi were enriched in either astrocytes and/or microglia and these changes might reflect a glia contamination. Yet *Sik1* (changing in CamKIIa cortical neurons) and *Cyp2s1* and *Oprm1* (both changing in PV neurons) were highest expressed in neurons, and it is possible that even such few changes are sufficient to induce neuronal dysfunction and symptoms.

## Different cell types display distinct PrD-related changes

We next focused on the terminal disease stage. Hierarchical clustering based on Euclidean distances and a PCA revealed a segregation of the terminal samples according to both cell type and treatment (*Figure 5a–b*). We observed a low correlation between the translational changes in the different cell types (*Figure 5—figure supplement 1a*) and found that most prion-induced changes are cell-type-specific (*Figure 5c* and *Supplementary file 9*). Yet, five genes showed an increase in translation in

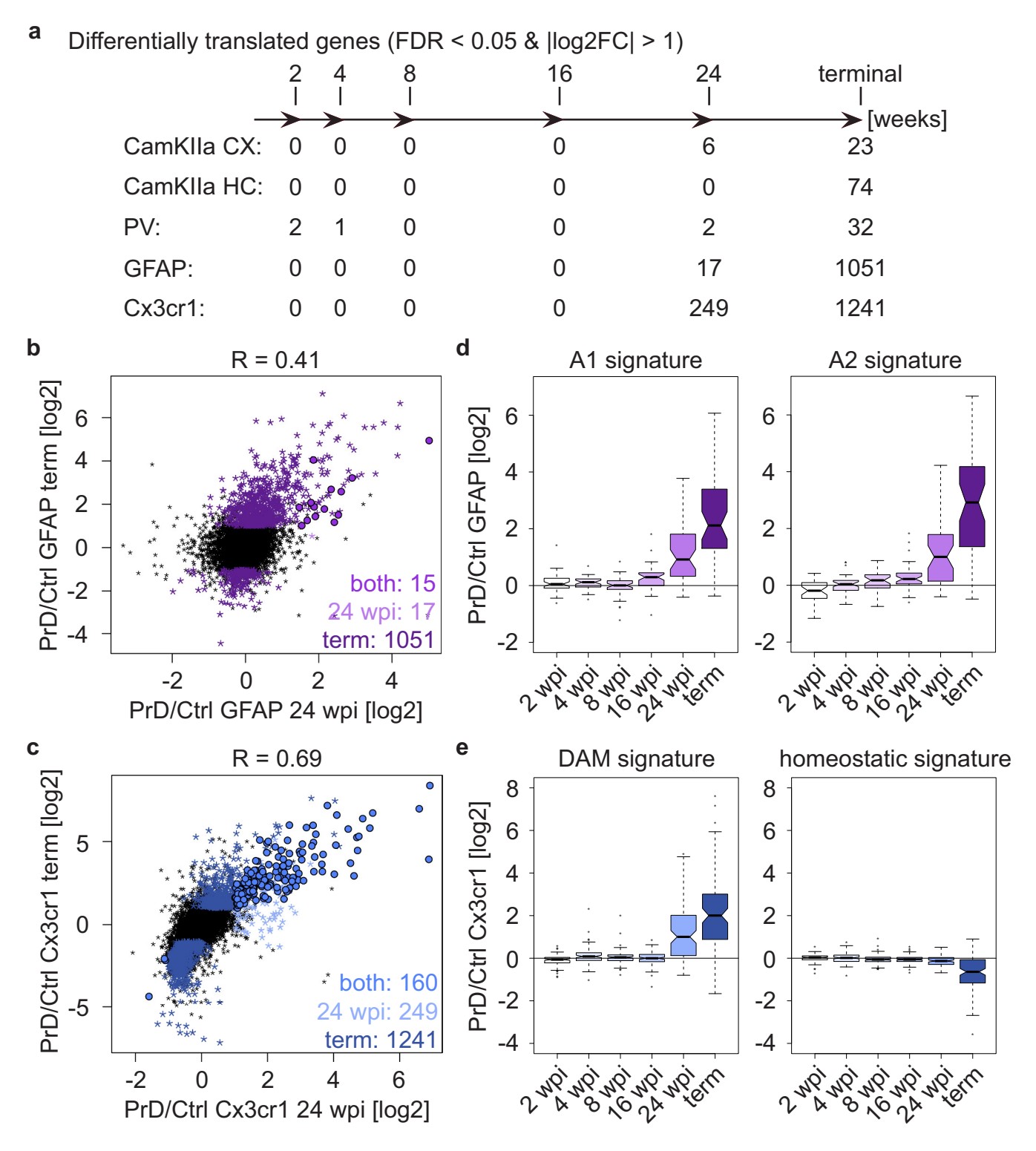

**Figure 4.** Cell-type-specific changes become only evident at late prion disease stages. (a) Schematic depicting numbers of differentially translated genes (DTGs: |log2FC| > 1 and FDR < 0.05) in the indicated cell types at different stages during disease progression (in the corresponding *Figure 4— figure supplement 2* all genes with an FDR < 0.05 are colored, independent of their log2FC). (b) Scatterplot comparing prion-induced changes in GFAP[+] cells at the pre-terminal (24 weeks post-inoculation (wpi)) and terminal time points. Genes changing at one or both time points are indicated. (c) Scatterplot comparing prion-induced changes in Cx3cr1[+] cells at the pre-terminal (24 wpi) and terminal time points. Genes changing at one or both time points are indicated. (d) Boxplots displaying translational changes of neurotoxic (A1 signature) and neuroprotective (A2 signature) genes in GFAP[+]

*Figure 4 continued on next page*

*Figure 4 continued*

cells during prion disease (PrD) progression. (**e**) Boxplots displaying translational changes of disease-associated microglia (DAM) and homeostatic microglia genes in Cx3cr1[+] cells during PrD progression.

The online version of this article includes the following figure supplement(s) for figure 4:

**Figure supplement 1.** Principal component analysis of control and prion disease samples.
**Figure supplement 2.** Identification of differentially translated transcripts in different cell types.
**Figure supplement 3.** Comparison of prion-induced changes during disease progression.

all analyzed cell types: *Gfap*, *Serpina3n*, *Apod*, *Mt1*, and *Lgals3bp* (*Figure 5—figure supplement 1b*). All five genes are highly translated in astrocytes, microglia or both (*Figure 5—figure supplement 1b* and *Supplementary file 9*), indicating that the increase in neuronal translation of these common genes is likely a contamination and reflects the pronounced astro- and microgliosis at the terminal PrD stage.

Interestingly, we detected pronounced regional differences in translation within one and the same cell type. The CamKIIa changes between cortex and hippocampus did not correlate (*Figure 5d*) and, apart from the common genes discussed above, only one additional gene was shared between the CamKIIa datasets. Similarly, most glia changes were specific to either astrocytes or microglia (*Figure 5e*). Yet, we also identified 170 genes that changed in both glia cell types, with most of them (n = 153) changing in the same direction. We analyzed the gene ontology (GO) terms associated with genes that increased specifically in astrocytes (n = 692), microglia (n = 442) or in both cell types (n = 136). We found an enrichment of immune response genes among all three groups of genes and, among astrocyte-specific increasing genes, an enrichment of terms related to cell adhesion and extracellular matrix organization, indicating that these processes might be affected in astrocytes (*Figure 5f* and *Supplementary file 10*). In contrast, many decreasing genes that are astrocyte-specific (n = 184) or shared (n = 17) have been linked to sterol and cholesterol biosynthesis, respectively, whereas microglia-specific decreasing genes (n = 640) were enriched for neuronal terms including synaptic transmission (*Figure 5f* and *Supplementary file 10*).

As stated earlier, microglia proliferation and neuronal loss are hallmarks of PrD, complicating the interpretation of microglia-specific decreasing genes. To address if some microglia changes are an artefact, we compared the translational change in microglia to the translational enrichment in different cell types compared to microglia (*Figure 5—figure supplement 1c*). Indeed, we observed a negative correlation between the translational change in microglia and neuronal enrichment: many microglia-specific decreasing genes (colored in blue) were strongly enriched in neuronal cell types (*Figure 5—figure supplement 1c*: first three panels). We did not observe such a relationship for genes that were enriched in astrocytes compared to microglia (*Figure 5—figure supplement 1c*: fourth panel) nor for genes that changed in astrocytes (*Figure 5—figure supplement 1d*). The decrease in neuronal-enriched genes specifically in microglia is thus likely an artefact and reflects a change in the composition of analyzed cell types, namely the increase in microglia relative to neurons. We believe that astrocyte-enriched genes are less affected by an increase in microglia since astrocytes are a lot more abundant than specific neuronal subtypes. We therefore only consider microglia-specific decreasing genes that are enriched in neurons to be an artifact and have flagged these genes in *Supplementary file 9*.

## Cell-type-specific ribosome profiling identifies novel PrD-induced changes

We previously published an extensive database of RNA expression changes in the hippocampus upon intracerebral and intraperitoneal prion inoculation (*Sorce et al., 2020*), and intersected the intraperitoneal RNAseq datasets with the cell-type-specific ribosome profiling data (*Supplementary files 9* and *11*). While we did not observe a correlation between mRNA abundance and cell-type-specific translational changes at 8 wpi (*Figure 6—figure supplement 1a*), the datasets correlated at the terminal stage (*Figure 6—figure supplement 1b*), and many genes changed in both mRNA abundance and cell-type-specific translation (*Figure 6a*).

We previously reported that terminal mRNA abundance changes correspond to genes enriched in microglia and astrocytes. Consistently, we observed that microglia PTGs showed a strong increase

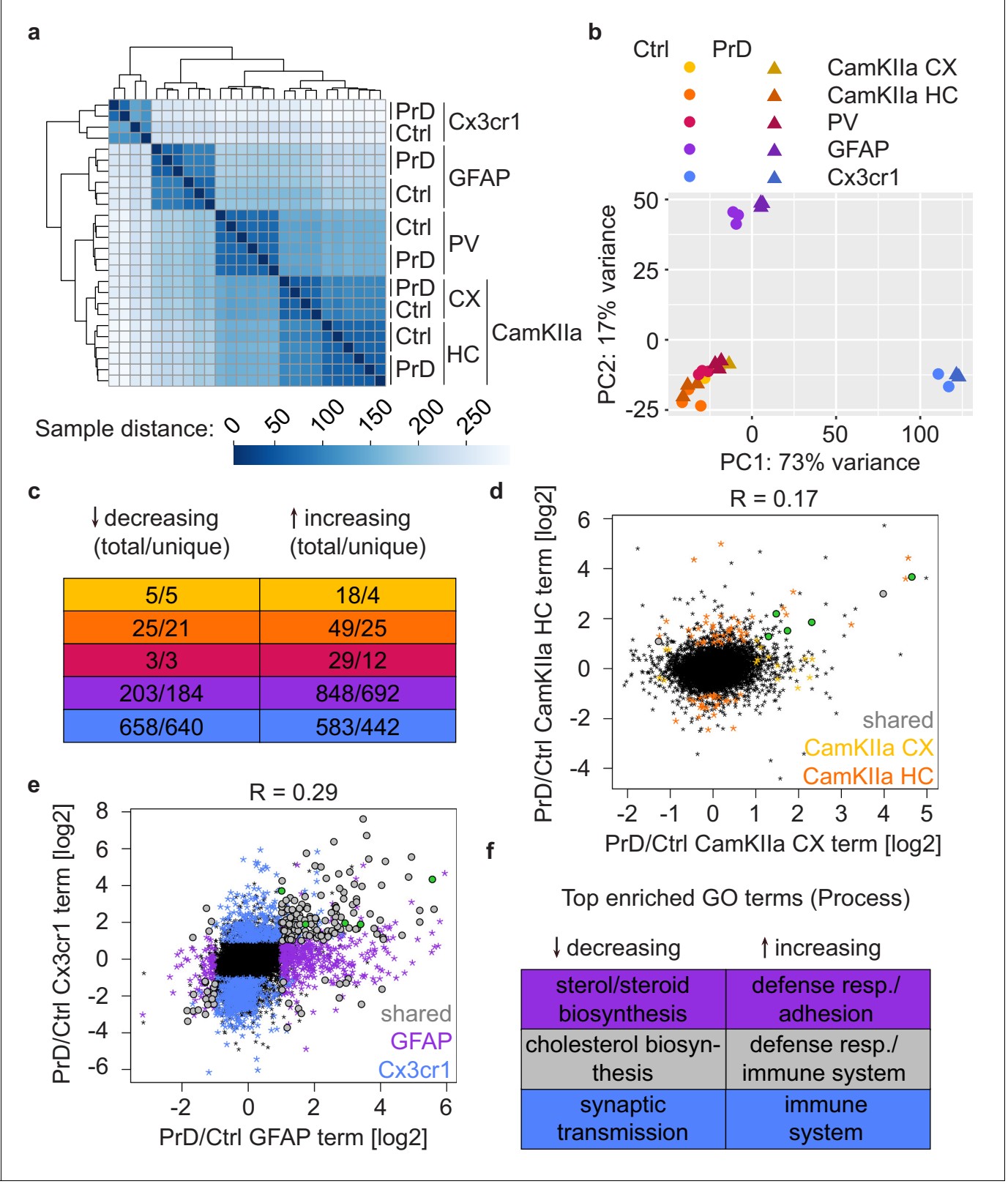

**Figure 5.** Different cell types display distinct prion disease related changes. (**a**) Hierarchical clustering of control and PrD samples at the terminal stage based on Euclidean distances. Heatmaps depicting the sample distance at each time point based on rlog transformed RPF counts per gene show a clustering of samples according to treatment and Cre driver (~cell type). (**b**) Principal component analysis of control (ctrl) and prion disease (PrD)

*Figure 5 continued on next page*

*Figure 5 continued*

samples at the terminal stage based on rlog transformed RPF counts per gene revealing a separation according to Cre driver (~cell type). (**c**) Table showing the number of total and cell-type-specific (unique) changes in the different cell types at the terminal stage. (**d**) Scatterplot comparing prion-induced changes between cortical and hippocampal CamKIIa[+] cells at the terminal time point. Genes changing in one or both cell types are indicated. Genes that change in all cell types are shown in green. (**e**) Scatterplot comparing prion-induced changes in GFAP- and Cx3cr1[+] cells at the terminal time point. Genes changing in one or both cell types are indicated. Genes that change in all cell types are shown in green. (**f**) Table displaying top enriched GO terms among genes changing only in GFAP[+] cells, only in Cx3cr1[+] cells or in both glia cell types.

The online version of this article includes the following figure supplement(s) for figure 5:

**Figure supplement 1.** Identification of common and cell-type-specific translational changes.

in the RNAseq dataset (*Figure 6b*), and that more than a third of the genes changing in mRNA abundance are microglia PTGs (566 out of 1451), most of which specifically changed in the RNAseq dataset (*Figure 6a*, n = 411). The RNAseq dataset showed the highest correlation with GFAP translational changes (*Figure 6c* and *Figure 6—figure supplement 1b*), which is consistent with astrocytes being the most abundant cell type. Yet, many genes that increased in mRNA abundance were enriched in microglia (*Figure 6d*), and thus reflect microglia proliferation rather than an increase in microglia-gene expression on a cellular level. We specifically focused on four genes that were enriched in astrocytes (*Glul* and *Gfap*) and microglia (*Aif1* encoding for IBA1, and *Mmp9*). While we observed no change in cell-type-specific translation nor protein expression in whole brain extracts for *Glul*, the protein expression and the translation in all assessed cell types strongly increased for *Gfap* (*Figure 6e–f* and *Figure 6—figure supplement 2a*). This indicates that the number of astrocytes does not change during PrD, yet specific genes such as *Gfap* are strongly upregulated in astrocytes during PrD progression. In contrast, we observed an increase in both IBA1 and MMP9 protein expression, whereas *Aif1* translation did not change and *Mmp9* translation even decreased, specifically in microglia (*Figure 6e–f* and *Figure 6—figure supplement 2a*). The increased microglia proliferation therefore not only accounts for the increase in IBA1 protein levels (no cell-type specific change in translation was detectable), but also additionally masks the microglia-specific decrease in *Mmp9* translation. We additionally assessed NeuN expression in hippocampal and cortical lysates as well as PV expression in whole brain lysates but did not observe a decrease in neuronal markers in PrD (*Figure 6g* and *Figure 6—figure supplement 2a*). These results support our initial observation (*Figure 1* and *Figure 1—figure supplement 1*) that terminally diseased PrD mice showed no or only little neuronal loss and explain why neuronal enriched genes show no or only a minor decrease in the RNAseq dataset (*Figure 6a*).

Since terminally sick PrD mice show an upregulation of GFAP and an increase in microglia, we determined the levels of Rpl10a:GFP in mice bred with GFAP Cre and Cx3cr1 Cre lines. promoters. GFP levels remained constant in GFAP Cre/Rpl10a:GFP mice (*Figure 6—figure supplement 2b*), indicating that an increased GFAP promoter activity does not lead to additional Cre recombination events. In contrast, GFP levels increased in Cx3cr1 Cre/Rpl10a:GFP mice (*Figure 6—figure supplement 2c*), most likely reflecting the increase in microglia numbers. These results are consistent with our ribosome profiling data and suggests that a difference in GFP expression does not impact our analyses. Collectively, our data highlights the importance of studying individual cell types, allowing us to identify ~1500 novel genes that showed PrD-dependent translational changes, which were missed with more conventional approaches such as RNAseq of brain regions.

## Discussion

Genome-wide prion-induced transcriptional changes were previously investigated in large brain regions through RNAseq (*Kanata et al., 2019*; *Sorce et al., 2020*). While these analyses can identify prominent changes in RNA expression, splicing and editing, they do not possess the resolution necessary to identify cell-type specific changes. Moreover, the assessment of entire brain regions cannot discriminate between RNA expression changes within one cell type and an altered cellular composition. A recent report described prion-induced molecular changes in neuronal-enriched microdissected regions in the hippocampus and cerebellum via microarrays and identified changes linked to neuronal function (*Majer et al., 2019*). Yet, microdissection can fail to capture events occurring within neuronal processes or synapses and, crucially, may induce gene expression changes during

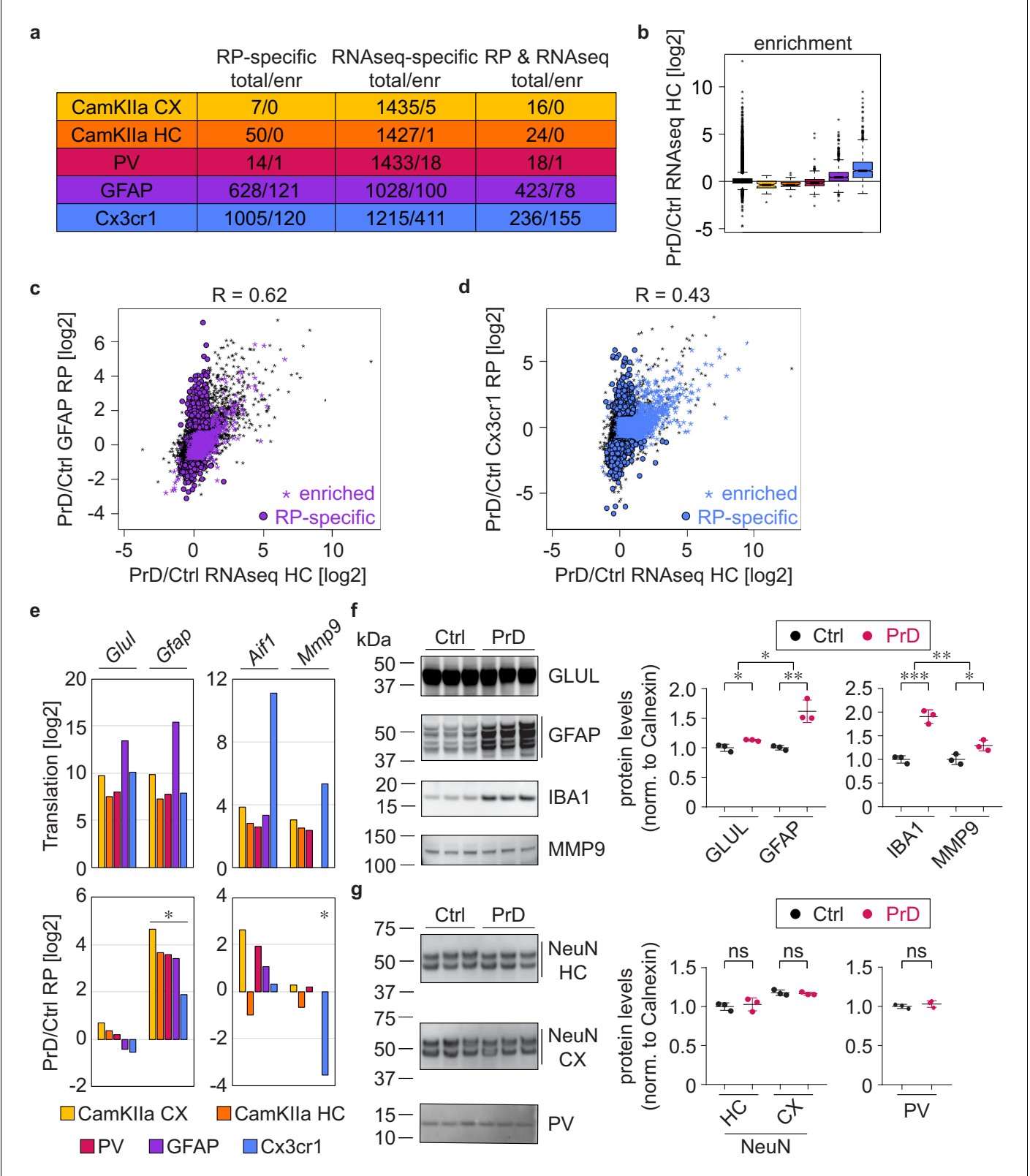

**Figure 6.** Cell-type-specific ribosome profiling identifies novel prion disease induced changes. (a) Table displaying the number of changes in the different cell types at the terminal stage and how many of them are preferentially translated genes (PTGs). Shown are numbers of genes that change either in the ribosome profiling (RP) or the RNAseq or in both datasets. (d) Boxplots displaying terminal RNA expression changes in the hippocampus of all expressed genes and PTGs. (c) Scatterplot comparing prion-induced RNA expression changes in the hippocampus and translational changes in

*Figure 6 continued on next page*

*Figure 6 continued*

GFAP⁺ cells. GFAP PTGs and genes that change only in the RP but not the RNAseq dataset are indicated. (**d**) Scatterplot comparing prion-induced RNA expression changes in the hippocampus and translational changes in Cx3cr1⁺ cells. Cx3cr1 PTGs and genes that change only in the RP but not the RNAseq dataset are indicated. (**e**) Bars represent the translation (log2 average of normalized counts per gene; DESeq2 output) and translational change (log2FC; DESeq2 output) in the different cell types. Shown are genes that are enriched in astrocytes (*Glul* and *Gfap*) and microglia (*Aif1* encoding for IBA1, and *Mmp9*). Significant changes (Benjamini Hochberg adjusted p value < 0.05; derived from DESeq2 analyses) are marked with an asterisk. (**f**) Western blot and its quantification showing protein levels of astrocyte (GLUL and GFAP) and microglia (IBA1 and MMP9) enriched genes in control (ctrl) and terminal prion disease (PrD) samples (*p<0.05; **p<0.01; ***p<0.001; two-tailed t test comparing terminal PrD vs ctrl samples; error bars: standard deviation). (**g**) Western blot and its quantification showing neuronal protein levels (NeuN in cortex and hippocampus, PV in whole brain) in ctrl and terminal PrD samples (ns = not significant; two-tailed t test comparing terminal PrD vs ctrl samples; error bars: standard deviation).

The online version of this article includes the following figure supplement(s) for figure 6:

**Figure supplement 1.** Comparison of prion-induced changes in RNA expression and translation.

**Figure supplement 2.** Comparison of GFP expression.

the isolation procedure. Additionally, any differences in mRNA abundance do not necessarily translate into altered protein expression. In fact post-transcriptional regulation seems to be particularly important in the nervous system, and numerous RNA-binding proteins are expressed in a neuron-specific manner (*Darnell, 2013*), the dysregulation of mRNA translation has been linked to multiple neurodegenerative diseases (*Kapur et al., 2017*), and an activation of the unfolded protein response (UPR) in PrD mice has been shown to specifically affect the translation rather than the expression of RNAs (*Moreno et al., 2012*).

Here, we have addressed these limitations by investigating PrD-relevant cell types throughout disease progression. We have identified preferentially translated genes (PTGs) and prion-induced changes in the analyzed cell types. We assessed astrocytes and microglia due to their activation being characteristics of PrD, PV neurons because they have previously been shown to be severely affected in PrD (*Ferrer et al., 1993*; *Guentchev et al., 1998*), and CamKIIa-positive pyramidal neurons in the cortex and hippocampus as major glutamatergic neuronal subtypes.

Differently from single-cell sequencing, cell-type-specific ribosome profiling assesses mRNA translation rather than expression, allows the identification of genome-wide changes and not only changes of highly expressed genes, and does not require manipulations likely to induce artefacts. One caveat of our approach lies within the unspecific pulldown of highly expressed genes, especially if they are expressed in an abundant cell type. Yet only 5 out of 2185 terminal changes changed in all 5 investigated cell types, indicating that the vast majority of changes are detectable in one or few cell types. In contrast, many neuronal-enriched genes appeared to decrease specifically in microglia. The latter observation most likely reflects a contamination of neuronal genes combined with an altered cellular composition (fewer neurons compared to microglia are present at the terminal stage). Importantly, our data allows us to identify such situations and take them into account when interpreting quantitative data. Collectively, our approach enables the assessment of biologically relevant cell-type-specific translation in a genome-wide manner, thereby complementing transcriptomic approaches.

The murine PrD model assessed here did not show a major decrease in neurons at the terminal stage. While this observation is consistent with previous reports, they are in stark contrast to the pathological changes characteristic for human PrD patients. Compared to humans, whose lives can be somewhat prolonged due to medical interventions, mice must be euthanized at a relatively early endpoint, well ahead of terminal disease, in order to comply with animal-welfare regulations. While these rules may have some justification, they render it legally impossible to investigate the true endpoint of mouse prion models. Consequently, the difference in neuronal loss is likely to reflect a difference in the disease stage. Interestingly, we observed only minor translational changes in different neuronal populations, although terminal PrD mice displayed pronounced neurological symptoms including piloerection, hind limb clasping, kyphosis and ataxia. It is possible that even few translational changes in neurons suffice to induce neurological dysfunction and symptoms. Additionally, our approach did not allow us to assess the impact of non-translated RNAs. Alternatively, neurological changes are evoked by subtle differences affecting neuronal connectivity, which may be induced by the widespread glia alterations described above but may escape translational analyses.

Translation of mRNA has previously been shown to be affected in a PrD mouse model which over-expresses PrP and are thus characterized by an accelerated PrD progression with pronounced loss of synapses and neurons (*Moreno et al., 2012*). These mice show a protracted UPR activation leading to increased phosphorylation of PERK and eIF2α, a global decrease in translation (including of *Actb*, *Snap25*, and *Psd95*), as well as increased ATF4/CHOP expression and apoptosis. While it remains unclear in which cell types these changes occur, neuronal overexpression of GADD34 globally decreased eIF2α phosphorylation in hippocampal extracts, rescued the reduced translation and neuronal loss, and increased survival. More recently, the same group could show that also astrocytic overexpression of GADD34 was neuroprotective and increased survival (*Smith et al., 2020*), highlighting the importance of astrocytes in neurodegeneration. Interestingly, in our PrD mouse model none of the analyzed cell types showed a differential translation of *Actb*, *Snap25*, *Psd95* (*Dlg4*), or *Chop* and only microglia showed an increased translation of *Atf4*. In addition, while Moreno et al. did not observe an increased expression of GADD34 in hippocampal extracts, we observed an increase in Gadd34 translation specifically in astrocytes, presumably leading to a reduced UPR. Considering the differences in UPR marker expression and neuronal loss between the different PrD mouse models, it is plausible that a reduced UPR activation in our mouse model might contribute to a less pronounced neurodegeneration.

Previously identified glia signatures increased in our dataset, including disease associated microglia (DAM) genes as well as neurotoxic (A1) astrocyte genes. In contrast, homeostatic microglia genes decreased during PrD progression. Neurotoxic A1 astrocytes are induced by activated microglia and are highly abundant in several neurodegenerative diseases (*Liddelow et al., 2017*). Yet, while preventing A1 formation after acute CNS injury reduced the loss of axotomized neurons (*Liddelow et al., 2017*), A1-astrocyte ablation in prion-inoculated mice accelerated disease progression (*Hartmann et al., 2019*), indicates that A1 astrocytes in PrD could be neuroprotective. Combined with our observations that neurotrophic (A2) astrocyte genes and Gadd34 increase in astrocytes, these results suggest that astrocytes might play a central role in limiting prion toxicity.

## Materials and methods

### Mice

Animal experiments were performed in compliance with the Swiss Animal Protection Law, under the approval of the Veterinary office of the Canton Zurich (animal permits ZH040/15, ZH139/16). Mice were kept in a conventional hygienic grade facility, constantly monitored by a sentinel program aimed at screening the presence of all bacterial, parasitic, and viral pathogens listed in the Federation of European Laboratory Animal Associations (FELASA). The light/dark cycle consisted of 12/12 hr with artificial light (40 Lux in the cage) from 07:00 to 19:00 hr. The temperature in the room was $21 \pm 1°C$, with a relative humidity of $50 \pm 5\%$. The air pressure was controlled at 50 Pa, with 15 complete changes of filtered air per hour (HEPA H 14 filter; Vokes-Air, Uster, Switzerland). Up to five mice were housed in IVC type II long cages with autoclaved dust-free Lignocel SELECT Premium Hygiene Einstreu (140–160 g/cage) (J. Rettenmaier and Söhne GmbH), autoclaved $20 \times 21$ cm paper tissues (Zellstoff), autoclaved hay and a carton refuge mouse hut as nesting material. Individual housing was avoided, and all efforts were made to prevent or minimize animal discomfort and suffering. Prion-inoculated and control-injected mice were regularly monitored for the development of clinical signs and humane termination criteria were employed.

The following mouse strains were used in this study: Rpl10a:GFP (RRID:IMSR_JAX:022367), Camk2a Cre (RRID:IMSR_JAX:005359), Cx3cr1$^{CreER}$ (RRID:IMSR_JAX:021160), GFAP Cre (RRID: IMSR_JAX:024098), and Pvalb Cre (MGI: 3798581). To induce Cre recombinase expression, 3-week-old Cx3cr1$^{CreER}$ mice received Tamoxifen chow (400 mg/kg; TD.55125.I) for 4 weeks. Eleven-week-old mice were intraperitoneally injected with 100 µl of RML6 prions (passage 6 of Rocky Mountain Laboratory strain mouse-adapted scrapie prions) containing 8.02 log $LD_{50}$ of infectious units per ml. Control inoculations were performed using 100 µl of non-infectious brain homogenate (NBH) from CD-1 mice at the same dilution. After inoculation, mice were initially monitored three times per week. After clinical onset, mice were monitored daily. Mice were sacrificed at pre-defined time points (corresponding to an age of 13 to 35 weeks) at the same time of the day. Prion-inoculated mice allocated to the terminal group were sacrificed upon clear signs of terminal PrD including

piloerection, hind limb clasping, kyphosis and ataxia (corresponding to an age of 42 to 43 weeks). This time point corresponds to the last time point when mice can be humanely euthanized and is reached 31–32 weeks post -ntraperitoneal prion inoculation. Control-injected mice assigned to the latest time point group were sacrificed at the same time as terminally ill mice.

Mice were sacrificed by carbon-dioxide-induced deep anesthesia followed by decapitation, and brains were immediately dissected. For ribosome profiling, brain regions (cortex and hippocampus) and hemispheres were dissected and flash frozen in liquid nitrogen. For immunohistochemistry, whole brains were fixed overnight in formalin at 4 ˚C, and prion-decontaminated at RT (1 hr in 100% formic acid, followed by a post-fixation step of 2 hr in formalin) before being paraffin embedded.

## Immunohistochemistry

Histological analyses were performed on 2-µm-thick sections from formalin fixed, formic acid treated, paraffin-embedded brain tissues. Sections were subjected to deparaffinization through graded alcohols, followed by heat-induced antigen retrieval performed in 10 mM citrate buffer (pH6). Stainings were performed with the following antibodies: IBA1 (1:125, Wako, 019–1974, RRID: AB_839504), GFAP (1:250, Dako, Z0334, RRID:AB_10013382), GFP (1:250, Rockland Immunochemicals, 600-101-215, RRID:AB_218182), NeuN (1:250, Merck Millipore, ABN78, RRID:AB_10807945), PV (1:500, Sigma Aldrich, P3088, RRID:AB_477329) and CamKIIa (1:100, Santa Cruz Biotechnology, sc-5306, RRID:AB_626788). The following secondary antibodies were used: anti-goat Alexa 488 IgG (H+L) (1:1000, A11055, Invitrogen, RRID:AB_2534102), anti-mouse Alexa 555 IgG (H+L) (1:1000, A21422, Invitrogen, RRID:AB_2535844) and anti-rabbit Alexa 555 IgG (H+L) (1:1000, A31572, Invitrogen, RRID:AB_162543). Images were acquired using the Olympus BX61 Upright Fluorescence Microscope.

## Western blot analysis

Samples were lysed in 1 ml cell-lysis buffer (20 mM Hepes-KOH, pH 7.4, 150 mM KCl, 5 mM $MgCl_2$, 1% IGEPAL) supplemented with protease inhibitor cocktail (Roche 11873580001), homogenized at 5000 rpm (15 s) with a Precellys24 Sample Homogenizer (LABGENE Scientific SA, BER300P24), incubated on ice for 20 min and cleared by centrifugation at 20,000 g, 4˚ C for 10 min in a tabletop centrifuge (Eppendorf 5417 R). Whole protein concentrations were measured with a BCA assay (Thermo Scientific). Samples were boiled in 4 x LDS (Invitrogen, supplemented with 10 mM DTT) at 95˚C for 10 min. Of total protein per sample, 20 µg were loaded on a gradient of 4–12% Novex Bis-Tris Gel (Invitrogen) for electrophoresis at 80 V for 15 min, followed by constant voltage of 160 V. After transfer to PVDF or Nitrocellulose membranes with the iBlot system (Life Technologies), the membranes were blocked with 5% Sureblock (LubioScience) in PBS-T (PBS + 0.2% Tween-20) for 1 hr at room temperature. Primary antibodies were incubated overnight in PBS-T with 5% Sureblock at 4˚C. The membranes were washed thrice with PBS-T for 10 min before incubating with secondary antibodies coupled to horseradish peroxidase for 1 hr at room temperature. Membranes were washed thrice with PBS-T for 10 min and developed with a Crescendo chemiluminescence substrate system (Millipore). Signal was detected using a LAS-3000 Luminescent Image Analyzer (Fujifilm), and analyzed with Quantity One (Bio-Rad, RRID:SCR_014280). The same membranes were then reprobed by incubating them for 30 min at room temperature in Restore Western Blot Stripping Buffer (Thermo Fischer Scientific), followed by washing thrice with PBS-T for 10 min and overnight incubation with the primary antibody anti-Calnexin (1:2000, Enzo Life Sciences, ADI-SPA-865, RRID:AB_10618434) at 4˚C. Secondary antibody incubation, development and detection were performed as described above.

The following primary antibodies were used for western blot analysis: anti-IBA1 (1:1000, Wako, 019–1974, RRID:AB_839504), anti-GFAP (1:1000, Dako, Z0334, RRID:AB_10013382), anti-GFP (1:1000, Rockland Immunochemicals, 600-101-215, RRID:AB_218182), anti-Calnexin (1:2000, Enzo Life Sciences, ADI-SPA-865, RRID:AB_10618434), anti-NeuN (1:2000, Merck Millipore, ABN78, RRID: AB_10807945), anti-MMP9 (1:1000, Abcam, ab38898, RRID:AB_776512), anti-Glutamine Synthetase (1:1000, Abcam, ab176562, RRID:AB_2868472) and anti-PV (1:500, Swant, PV27, RRID:AB_2631173). The following secondary antibodies were used: HRP-tagged goat anti-rabbit IgG (H+L) (1:3000, 111.035.045, Jackson ImmunoResearch, RRID:AB_2337938) and HRP-tagged donkey anti-goat IgG (H+L) (1:3000, Jackson ImmunoResearch, 705-035-147, RRID:AB_2313587).

## Ribosome profiling

Samples were lysed at 5000 rpm (15 s) with a Precellys24 Sample Homogenizer (LABGENE Scientific SA, BER300P24) in 1 ml lysis buffer (20 mM Hepes-KOH, pH 7.4, 150 mM KCl, 5 mM MgCl$_2$, 1% IGE-PAL, 0.5 mM DTT, 40 U/ml RNAsin Plus (Promega N2615), 100 µg/ml cycloheximide, one Complete EDTA-Free Protease Inhibitor Cocktail tablet (Roche 11873580001) per 10 ml). Lysates were incubated for 30 min on ice and centrifuged at 20,000 g (10 min at 4°C) before being subjected to partial RNA digestion via 45 min incubation at RT with CaCl$_2$ (1 mM final) and 300U micrococcal nuclease (0.3 U/ml; Thermo Scientific EN0181). The reaction was stopped by placing samples on ice and adding EGTA (10 mM final). GFP-tagged ribosomes were immunoprecipitated with a 1:1 mixture of monoclonal GFP antibodies (Memorial Sloan Kettering Monoclonal Antibody Facility, 19F7 and 19C8, RRID:AB_2716736 and RRID:AB_2716737). 200 µl protein G beads (Invitrogen 10004D), were coupled with 25 µg of each antibody, incubated with brain lysates for 1 hr at 4°C and washed thrice with wash buffer (50 mM Tris, pH7.5, 500 mM KCl, 5 mM MgCl$_2$, 1% IGEPAL, 0.5 mM DTT, 40 U/ml RNAsin Plus, 100 µg/ml cycloheximide). RPFs were isolated by incubating beads for 15 min in release buffer (20 mM Hepes-KOH, pH 7.4, 100 mM NaCl, 30 mM EDTA, 0.5 mM DTT, 40 U/ml RNAsin Plus), addition of NaCl (0.5M final) and 10 min incubation, and ultracentrifugation at 200,000 g (2 hr at 4 °C). The RPF-containing supernatant was run over G25 columns (GE Healthcare 95017–621) and incubated with 4U alkaline phosphatase (Roche 10713023001) to remove 3' phosphates. RPFs were extracted with Trizol LS (Invitrogen 10296–028) according to the manufacturer's recommendations and subjected to ligation (2 hr at RT) with a pre-adenylated 3' linker (2 µM final) and a truncated T4 ligase (NEB M0373L). Ligated RPFs of 50–70 nucleotides (nt) were size selected by Urea PAGE electrophoresis and eluted by incubating crushed gel pieces at 1200 rpm (45 min at 50 °C) with 1M NaOAc (pH 5.2) and 1 mM EDTA. Gel pieces were removed with filter tubes (Corning 8160), and RNA was precipitated before being reverse transcribed with SSIII (Invitrogen 18080–044) in the presence of BrdUTP (Sigma B0631, RRID:AB_306886). RT primers and the cloning and isolation of BrdUTP-labeled footprints have previously been described (Hwang et al., 2016; Weyn-Vanhentenryck et al., 2014). Briefly, the RT primer contains a 14-nt degenerate linker (a 3-nt degenerate sequence, a 4-nt multiplexing index, and a 7-nt unique molecular identifier), a 5'linker for PCR amplification, a spacer to prevent rolling circle amplification after circularization, and the reverse-complementary sequence of the 3'linker for reverse transcription. BrdUTP-labeled cDNA was specifically isolated via two sequential BrdUTP immunoprecipitations (with Abcam AB8955) and circularized with CircLigase II (Epicentre CL9025K). RPF libraries were PCR amplified with Phusion High-Fidelity DNA Polymerase (NEB M0530A) and the primer set DP5-PE/SP3-PE. The addition of SYBR Green (Invitrogen S7563) to the PCR reaction allowed us to track and stop library amplifications as soon as sufficient DNA had been generated to prevent overamplification-related PCR artifacts. Ribosome profiling libraries were isolated with AMPure XP beads (Agencourt A63880), quantified using the QIAxcel DNA High Resolution Kit for pooling. The quality and quantity of the pooled libraries were validated using Tapestation (Agilent, Waldbronn, Germany). The libraries were diluted to 10 nM in 10 mM Tris-Cl, pH 8.5 containing 0.1% Tween 20. The TruSeq SR Cluster Kit v4-cBot-HS (Illumina, Inc, California, USA) was used for cluster generation and sequencing was performed on a Illumina HiSeq 2500 single end 50 bp using the TruSeq SBS Kit v4-HS (Illumina, Inc, California, USA).

## Primer sequences

Pre-adenylated 3' linker:

    5rApp/GTGTCAGTCACTTCCAGCGG/3ddC

RT primer #1:

    /5Phos/DDDCGATNNNNNNNAGATCGGAAGAGCGTCGT/iSp18/CACTCA/iSp18/CCGC
    TGGAAGTGACTGAC

RT primer #2:

    /5Phos/DDDTAGCNNNNNNNAGATCGGAAGAGCGTCGT/iSp18/CACTCA/iSp18/CCGC
    TGGAAGTGACTGAC

RT primer #3:

/5Phos/DDDATCGNNNNNNNAGATCGGAAGAGCGTCGT/iSp18/CACTCA/iSp18/CCGC
TGGAAGTGACTGAC

RT primer #4:

/5Phos/DDDGCTANNNNNNNAGATCGGAAGAGCGTCGT/iSp18/CACTCA/iSp18/CCGC
TGGAAGTGACTGAC

RT primer #5:

/5Phos/DDDCTAGNNNNNNNAGATCGGAAGAGCGTCGT/iSp18/CACTCA/iSp18/CCGC
TGGAAGTGACTGAC

RT primer #6:

/5Phos/DDDGATCNNNNNNNAGATCGGAAGAGCGTCGT/iSp18/CACTCA/iSp18/CCGC
TGGAAGTGACTGAC

RT primer #7:

/5Phos/DDDAGCTNNNNNNNAGATCGGAAGAGCGTCGT/iSp18/CACTCA/iSp18/CCGC
TGGAAGTGACTGAC

RT primer #8:

/5Phos/DDDTCGANNNNNNNAGATCGGAAGAGCGTCGT/iSp18/CACTCA/iSp18/CCGC
TGGAAGTGACTGAC

DP5-PE:

AATGATACGGCGACCACCGAGATCTACACTCTTTCCCTACACGACGCTCTTCCGATCT

SP3-PE:

CAAGCAGAAGACGGCATACGAGATCTCGGCATTCCTGCCGCTGGAAGTGACTGACAC

## Data analysis

Ribosome profiling reads were processed with CTK (RRID:SCR_019034; *Shah et al., 2017*) and the FASTX-Toolkit (RRID:SCR_019035). Fastq files were de-multiplexed based on the 4-nt multiplexing index and filtered based on quality score (minimum of 20 from nt 1–14, mean of 20 from nt 15–40). The 3′ linker was removed, exact sequences were collapsed, and the 14-nt adaptor was trimmed. Reads were aligned to the GRCm38.p5 build of the mouse genome and uniquely aligned reads were summarized per gene using STAR v2.6.0c (RRID:SCR_015899; *Dobin et al., 2013*). STAR gene count output files were analyzed in R (www.r-project.org, RRID:SCR_001905; *Ihaka and Gentleman, 1996*), using Bioconductor (RRID:SCR_006442; *Gentleman et al., 2004*), and the packages 'DESeq2' (RRID:SCR_015687; *Love et al., 2014*) and 'ggplot2' (RRID:SCR_014601; *Wickham, 2016*). Data discussed in this manuscript has been deposited in the GEO database and is accessible through accession number GSE149805.

Gene Ontology analyses were performed with GORILLA (RRID:SCR_006848; http://cbl-gorilla.cs.technion.ac.il). We compared genes specifically decreasing (164 out of 184 were present in the GO database) or increasing (634 out of 692 were present) in astrocytes to astrocyte-expressed genes (12,888 out of 14,280 were present). Genes specifically decreasing (600 out of 640 were present in the GO database) or increasing (422 out of 442 were present) in microglia were compared to microglia-expressed genes (11,164 out of 11,919 were present), and genes decreasing (13 out of 17 were present) or increasing (126 out of 136 were present) in both astrocytes and microglia were compared to genes expressed in both cell types (13,329 out of 14,837 were present).

## Acknowledgements

We thank Catherine Aquino Fournier and the Functional Genomics Center Zurich for RNA sequencing, Mirzet Delic for excellent technical help, and Asvin Lakkaraju and members of the Aguzzi laboratory for critical comments and for discussions. AA is the recipient of an Advanced Grant of the

European Research Council, the Swiss National Foundation, the Clinical Research Priority Programs 'Small RNAs' and 'Human Hemato-Lymphatic Diseases', the Nomis Foundation and a donation from the estate of Dr. Hans Salvisberg. CS was supported by a Marie Curie Individual Fellowship.

## Additional information

### Funding

| Funder | Grant reference number | Author |
|---|---|---|
| H2020 Marie Skłodowska-Curie Actions | 706138 | Claudia Scheckel |
| H2020 European Research Council | 670958 | Adriano Aguzzi |
| Schweizerischer Nationalfonds zur Förderung der Wissenschaftlichen Forschung | 179040 | Adriano Aguzzi |
| Schweizerischer Nationalfonds zur Förderung der Wissenschaftlichen Forschung | 183563 | Adriano Aguzzi |
| NOMIS Foundation | Distinguished Scientist Award | Adriano Aguzzi |

The funders had no role in study design, data collection and interpretation, or the decision to submit the work for publication.

### Author contributions

Claudia Scheckel, Conceptualization, Data curation, Formal analysis, Supervision, Funding acquisition, Methodology, Writing - original draft, Project administration, Writing - review and editing; Marigona Imeri, Petra Schwarz, Data curation, Methodology; Adriano Aguzzi, Conceptualization, Supervision, Funding acquisition, Writing - original draft, Project administration, Writing - review and editing

### Author ORCIDs

Claudia Scheckel ⓘD https://orcid.org/0000-0002-1649-8486
Petra Schwarz ⓘD http://orcid.org/0000-0003-1686-8624
Adriano Aguzzi ⓘD https://orcid.org/0000-0002-0344-6708

### Ethics

Animal experimentation: Animal experiments were performed in compliance with the Swiss Animal Protection Law, under the approval of the Veterinary office of the Canton Zurich (animal permits ZH040/15, ZH139/16).

### Decision letter and Author response

Decision letter https://doi.org/10.7554/eLife.62911.sa1
Author response https://doi.org/10.7554/eLife.62911.sa2

## Additional files

### Supplementary files

• Supplementary file 1. Sample summary. Shown are sample information (sample name, analyzed region, gender, treatment, time point, Cre driver, and multiplex index), number of unique reads, and number and percentage of rRNA reads, aligned reads and aligned reads mapping to the coding sequence (CDS).

• Supplementary file 2. Identification of preferentially translated genes. Table displaying Ensemble GeneID, MGI Symbol, the individual and average translation (rlog transformed RPF counts per gene)

and difference analysis between all cell types including mean of normalized RPFs per gene (base-Mean), log2FC, pvalue and Benjamini-Hochberg adjusted p values (padj). The last column specifies preferentially translated genes (PTG), which are differentially translated (|log2FC| > 2 and padj <0.05) between one cell type versus all others.

• Supplementary file 3. Comparison of cell-type-specific translation and expression. Table including Ensemble GeneID, MGI Symbol, coding sequence (CDS) length, the individual and average translation (log2 transformed fpkm values) assessed by ribosome profiling (RP) and the average expression (log2 transformed fpkm values) assessed by ribotag (RT) in the indicated cell types. Counts per gene were normalized for sequencing depth and either CDS length (RP) or transcript length (RT). Additional columns specify if a gene is preferentially translated according to our dataset (column PTG), if it is one of the top 100 enriched genes according to *Zhang et al., 2014* (column Zhang) or if it was identified to be a marker gene according to *Hrvatin et al., 2018* (column Hrvatin).

• Supplementary file 4. Translational changes in CamKIIa cortical neurons during prion disease progression. Table displaying Ensemble GeneID, MGI Symbol, the individual and average translation (rlog transformed RPF counts per gene) and difference analysis between prion disease (PrD) vs control (ctrl) samples through disease progression in CamKIIa cortical neurons, including mean of normalized RPFs per gene (baseMean), log2FC, pvalue and Benjamini-Hochberg adjusted p values (padj).

• Supplementary file 5. Translational changes in CamKIIa hippocampal neurons during prion disease progression. Table displaying Ensemble GeneID, MGI Symbol, the individual and average translation (rlog transformed RPF counts per gene) and difference analysis between prion disease (PrD) vs control (ctrl) samples through disease progression in CamKIIa hippocampal neurons, including mean of normalized RPFs per gene (baseMean), log2FC, pvalue and Benjamini-Hochberg adjusted p values (padj).

• Supplementary file 6. Translational changes in PV neurons during prion disease progression. Table displaying Ensemble GeneID, MGI Symbol, the individual and average translation (rlog transformed RPF counts per gene) and difference analysis between prion disease (PrD) vs control (ctrl) samples through disease progression in PV neurons, including mean of normalized RPFs per gene (baseMean), log2FC, pvalue and Benjamini-Hochberg adjusted p values (padj).

• Supplementary file 7. Translational changes in GFAP[+] cells during prion disease progression. Table displaying Ensemble GeneID, MGI Symbol, the individual and average translation (rlog transformed RPF counts per gene) and difference analysis between prion disease (PrD) vs control (ctrl) samples through disease progression in GFAP[+] cells, including mean of normalized RPFs per gene (baseMean), log2FC, pvalue and Benjamini-Hochberg adjusted p values (padj). Genes associated with an A1 or an A2 signature are indicated in the last column.

• Supplementary file 8. Translational changes in Cx3cr1[+] cells during prion disease progression. Table displaying Ensemble GeneID, MGI Symbol, the individual and average translation (rlog transformed RPF counts per gene) and difference analysis between prion disease (PrD) vs control (ctrl) samples through disease progression in Cx3cr1[+] cells, including mean of normalized RPFs per gene (baseMean), log2FC, pvalue and Benjamini-Hochberg adjusted p values (padj). Genes associated with a homeostatic or disease-associated microglia (DAM) signature are indicated in the last column.

• Supplementary file 9. Summary table of all terminal changes. Table showing Ensemble GeneID, MGI Symbol, information of genes being preferentially translated (column PTG), and difference analysis between prion disease (PrD) vs control (ctrl) samples of cell-type-specific translation, and hippocampal RNA expression changes at the terminal stage intraperitoneally (ip) inoculated mice, including mean of normalized RPFs per gene (baseMean) for translation, and log2FC and Benjamini-Hochberg adjusted p values (padj) of translational and RNA expression changes. Additional columns specify if a gene is significantly (|log2FC| > 1 and FDR < 0.05) increasing or decreasing in a dataset and if decreasing Cx3Cr1-genes are neuronal enriched.

• Supplementary file 10. Gene Ontology analysis of terms related to biological processes. Genes decreasing and increasing in astrocytes, microglia and in both cell types were compared to respectively expressed genes.

• Supplementary file 11. Comparison of prion induced expression and cell-type-specific translational changes at the terminal stage. Table showing Ensemble GeneID, MGI Symbol, information of genes being preferentially translated (column PTG), and difference analysis between prion disease (PrD) vs control (ctrl) samples of cell-type-specific translation at the terminal stage, and hippocampal RNA expression changes at 8 weeks post-inoculation (wpi), and the terminal stage of intraperitoneally (ip) inoculated mice, including mean of normalized RPFs per gene (baseMean), log2FC, pvalue and Benjamini-Hochberg adjusted p values (padj).

• Transparent reporting form

## Data availability

Sequencing data has been deposited in GEO under accession code GSE149805.

The following dataset was generated:

| Author(s) | Year | Dataset title | Dataset URL | Database and Identifier |
|---|---|---|---|---|
| Scheckel C, Aguzzi A | 2020 | Cell-type specific translational changes in prion diseases | https://www.ncbi.nlm.nih.gov/geo/query/acc.cgi?acc=GSE149805 | NCBI Gene Expression Omnibus, GSE149805 |

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
