## [Decision Letter]

[Editors' note: this paper was reviewed by Review Commons.]

**Acceptance summary:**

This paper combines two complementary approaches, translating ribosome affinity purification (TRAP) and ribosome profiling, to provide the first quantitative insight into the impact of PrP^Sc^ on cell type-specific translation. Surprisingly few translational changes were detected in neurons. In contrast, substantial alterations to translation were evident in astrocytes and microglia prior to manifestation of prion disease features, suggesting that aberrant translation in these cell types may be a primary driver of neurodegeneration.

---

## [Author Response]

We thank the reviewers for the positive assessment of our work and their insightful remarks. Please find below a point-by-point response to each comment.

Reviewer #1 (Evidence, reproducibility and clarity):Scheckel et al. report a large dataset on cell type-specific translational profiling of PrD-associated molecular alterations in a mouse model thorough RiboTRAP and ribosome profiling approaches. They report a more severe alteration in the translatome specifically in astrocyte and microglia as compared to neuronal populations. This highlights that changes in these two cell classes might have a predominant role in the pathology of PrD.Data and the methods are presented such that they can be reproduced. The data analysis section of the manuscript could be further elaborated. In particular, it could be clarified which/how comparisons with existing dataset have been performed. Statistical analysis description is sometimes missing (e.g. Figure 6E, not clear what the stars on top of the bars stands for, which test was performed and the significance). Moreover, the section of the Materials and methods regarding the western blots presented in Figure 6 appear to be missing.

Figure 6E shows the output (log2 fold change) of DESeq2. Genes with a Benjamini-Hochberg adjusted p value < 0.05 (also derived from DESeq2) are marked with an asterisk. We have added this information to the legend, as well as methods regarding western blots.

Major concern:The most important improvement the authors should consider for their paper is to more specifically attempt to isolate specific effects on translational efficiency of mRNAs. As it stands, the authors largely use RiboTrap data as a reference to compare their footprinting data – but arguably, this misses mRNAs that are present in the transcriptome and not efficiently recruited onto ribosomes. It appears to be somewhat a lost opportunity to not attempt to test in the dataset (possibly by comparison to RNASeq from FACS isolated cells as a reference) whether there is a systematic change in translational efficiency (possibly in mRNAs with specific features?). In the current form, the RiboTrap and footprinting approaches largely serve to isolate mRNAs from Cre-defined cell types but given the lack of a "total transcriptome" reference from the respective cells, it cannot be easily interpreted whether certain transcripts are heavily regulated at the level of translation. Thus, despite using much more advanced methodologies than the Sorce study, the fundamental conclusions emerging from this work are rather similar to this previously published piece of work.

Translational changes can be assessed in a cell-type specific manner without artefacts related to dissociation/isolation procedures and are arguably more relevant than transcriptional changes (Haimon et al., 2018). Both, the assessment of translation as well as the investigation of specific cell types differentiates this study from transcriptional profiling studies including Sorce et al. Accordingly, our approach identified > 1000 cell-type specific translational changes that were missed in the Sorce study (Figure 5A-D).

We agree however with the reviewer that a comparison of our data with RiboTrap data does not take non-transcribed RNAs into account. We have refrained from such a comparison for several reasons:

1) We agree with the reviewer that a systematic comparison of transcriptomes and translatomes in the assessed cell types at every time point would have allowed us to identify genes regulated on a post-transcriptional level. The goal of this study was however to identify biologically relevant prion-induced molecular changes in a cell-type specific manner rather than identify post-transcriptional regulation. To assess the validity of our approach we chose closely related datasets (RiboTrap datasets) to compare our data to.

2) The inclusion of RNAseq datasets from FACS-isolated cells would require an additional 2 years of work since all samples and datasets would need to be newly generated (breeding mice, inoculating mice with prions and waiting for up to 8 months for mice to reach the terminal time point, establishing procedures, generating and analyzing datasets).

3) RNASeq from FACS isolated neurons is problematic due to neuronal processes often being lost during the dissociation/isolation procedures. Additionally, dissociation/isolation procedures typically introduce stress-related artefacts. These procedure-induced changes complicate comparisons with techniques that have been optimized to avoid such artefacts (including the method applied in this manuscript). Differences between transcriptional and translational datasets could thus be either due to post-transcriptional regulation or due to artefact differences and are likely difficult to interpret.

Additional suggestions:1) In Figure 1D the authors point out occasional neuronal cells exhibiting Rpl10a:GFP expression with arrows. It appears that these arrows may have moved during figure preparation – please check/fix if necessary.

Thank you for pointing this out. We have fixed the arrows.

2) In Figure 1—figure supplement 1B and C it appears that the PV labeling is missing in the panel for Rpl10a:GFP controls. If this is intentional, please indicate this in the figure legend.

A co-localization of GFP-positive cells and PV was assessed only in Cre-positive (GFP expressing) mice but not in Cre-negative mice that don’t express GFP. We have clarified this point in the corresponding figure legend.

3) It appears that the authors sequenced a significant number of libraries generated for multiple time points post-inoculation. From the figures and legends, it was not entirely clear to me, how many replicates were analyzed given that in some analyses samples from different time points were combined in a single plot.

All analyzed samples are listed in Supplementary file 1. We have emphasized this pointed in the Results section.

4) It was unclear to me how long after inoculation the group of "terminally ill" mice were sacrificed. Somewhere in the text it states that there are 2 months between 24 wpi and terminally ill – but it appears that this was not a preset timepoint but varied from animal to animal based on symptoms. Please clarify.

We sacrifice mice at the last humane time point possible at which they show terminal disease symptoms, including piloerection, hind limb clasping, kyphosis and ataxia. Intraperitoneal inoculated mice reach that time point at 31 – 32 weeks post inoculation (+/- few days). Control mice (inoculated with non-infectious brain homogenate) were sacrificed at the same time. We have clarified this point in the Material and methods section.

5) From the Western blot data in Figure 6F the authors conclude that GFAP expression is upregulated in PrD mice whereas astrocyte number is unchanged. Given that the translatome is assessed based on a Rpl10-GFP dependent on recombination mediated by Cre driven from GFAP promoter it is possible that the astrocytic alterations in ribosome footprints are in part a secondary consequence of increased Rpl10-GFP recombination/ expression in PrD mice (due to activation of the GFAP promoter). To estimate the impact of such an effect the authors should compare GFP levels in terminally ill control and PrD mice by western blotting.

We agree with the reviewer that this information would be important to add. We have therefore assessed GFP levels in Rpl10a:GFP mice bred with GFAP^Cre^ and Cx3cr1^CreER^ mice. The corresponding western blots are included in Figure 6—figure supplement 2. GFP levels remained constant in terminally ill GFAP^Cre^ mice. This is not surprising since even a low GFAP promoter activity is likely to allow sufficient Cre recombinase expression to remove a STOP cassette allowing GFP expression (controlled by the Rosa26 promoter) in GFAP^Cre^ mice.

In contrast, we observed an increase in GFP expression in terminally Cx3cr1^CreER^ mice, which is most likely linked to the increase in microglia numbers. As pointed out in the manuscript, the translational changes we identified cannot reflect differences in cell numbers due to the nature of our assay. This suggests that a difference in GFP expression does not impact our analyses.

We have added this data to the manuscript.

6) The western blot analysis of Figure 6F-G has been performed using a normalization over calnexin, yet no calnexin signals shown to support this statement.

We have included blots of the normalization control calnexin as Figure 6—figure supplement 2A.

7) Clarify the percentage of non-parenchimal machrophages that are accounting for the Cx3cr1^CreER^ mouse line since the authors consider this only to be a minor contamination.

The labeling of non-parenchymal macrophages using Cx3cr1^CreER^ mice has previously been estimated to be ~1% (Haimon et al., 2018). We have added this information to the manuscript.

8) Regarding the presentation of the data, Figure 5A would be clearer if in the y axes, for each cell type the order of PrD and Ctrl samples was maintained.

Figure 5A displays hierarchical clustering based on Euclidian distances. As samples are ordered according their distance from each other, we cannot change the order as suggested by the reviewer.

Reviewer #1 (Significance):Overall, this is an important and interesting study. Besides its insights into the biology, the transcriptomic data will provide a valuable resource for researchers in the field.Previous studies employed bulk RNAseq or microdissection for mapping transcriptomic changes (Majer et al, .2019; Sorce et al., 2020 and others). The Sorce et al. study concluded that astrocytic alterations in the transcriptome are more dominant than neuronal gene expression changes. While the conclusion of the present study remains the same, it is the first to use of ribosome profiling to dissect actively translated transcripts over the progression of the pathology in the mouse model. Thus, the data presented here would allow for identifying cell type-specific alterations as well as alterations specifically in mRNA translation which would be missed by bulk RNASeq and RNASeq on FACS-isolated cells. However, the authors do not fully capitalize on this strength, given that no detailed comparisons are done to a real transcriptome reference are performed (see above).This work is of broad interest to scientists in neurodegeneration as well as glial biology.Reviewer #2 (Evidence, reproducibility and clarity):Using a series of Cre-driven mouse strains a GFP-tagged version of RPL10a (a ribosomal protein) was targeted to different cell types allowing Dr Scheckel and colleagues to investigate translational changes as prion disease progresses in mice. Their data suggest massive changes in microglia and astrocytes but not neurons. The approach was particularly powerful as ribosome IP has been combined with ribosome profiling. The manuscript is very well written. What might help, however, is to make the figures more accessible (perhaps change some of the labelling?).I have only comments regarding some of the figures:Figure 1A: This scheme could be improved, adding wpi and better aligning the cell-types in relation to the time when the cell-types were analysed.

We have replaced weeks with wpi and changed the alignment of cell types to clarify that all cell types were analyzed at every time point.

Figure 1B-E: The resolution could be improved to better discern the different cell-types.

We submitted low-quality figures due to an upload limit but will submit final figures of higher quality. Additionally, we have added higher magnification pictures to better discern the different cell types as Figure 1—figure supplement 1D-E.

Figure 4: Astrocytes are categorised into A1 and A2 and microglia based on DAM and homeostatic signature (How does this relate to the M1 and M2 classification?).

The categorization of microglia into homeostatic and disease-associated (as well as other) microglia has largely replaced the initial categorization into pro-inflammatory M1 and anti-inflammatory M2 microglia (Dubbelaar et al., 2018), We have therefore opted for the more current categorization. This explanation has also been added to the manuscript.

Reviewer #3 (Evidence, reproducibility and clarity):Summary:The authors sampled actively translated proteins by cell type in the brains of RiboTag expressing mice under the control of cell specific Cre recombination to determine changes in the translational profiles. They injected prions IP to induce prion disease. Their model shows little to no neuron loss at the terminal stage due to animal welfare regulations, but neuronal loss is a key hallmark of prion disease, along with gliosis. However, since other groups under different animal welfare regulations have shown that prion injection is sufficient to fully model the disease given enough time, there is sufficient evidence that this model captures early disease pathogenesis.The methodology used here has some clear advantages over previous cell-type isolation methods that require more lengthy sorting procedures. However, proteins with a long half-life or tightly regulated levels (such as TDP-43) are likely underrepresented by this method. The method also depends strongly on the specificity of the Cre driver used; CamkIIa (excitatory N), parvalbumin (inhibitory N), GFAP (A), Cx3cr1 (microglia). While there is some off-target expression of the GFAP and Cx3cr1, the overall expression profiles generally match cell-specific transcriptomes obtained by other groups using other methods.They find major changes in astrocytes and microglia at terminal stages, after the onset of neurological symptoms, and comparatively fewer in neurons. Oligodendrocytes are not examined. The authors are commended on a thorough and well-designed study, especially in the comparison of multiple neuronal and glial types simultaneously.Major comments:Key conclusion 1: "Our results suggest that aberrant translation within glia may suffice to cause severe neurological symptoms and may even be the primary driver of prion disease." This conclusion is well-supported, serving as a hypothesis for future work.The data shows that the most abundant PTG changes are indeed in microglia at 24 wpi, before the onset of symptoms. In addition, although some genes are also differentially translated in the neuronal populations, examination of the Supplemental Tables shows that these are mostly highly expressed glial genes and could represent contamination of the sample during gliosis. The authors may wish to discuss this more prominently to avoid confusion. This data indeed suggests that glial changes alone are could be sufficient to produce the neurological symptoms in these mice. However, the authors should include discussion that the two genes changed at 24 weeks in PV neurons (Oprm1, Cyp2s1) do appear to be neuronal and may be relevant to pathogenesis as well. These mRNAs were also decreased in their previous paper conducting bulk sequencing in the hippocampus, according to the authors' online Prion RNAseq Database. Knockout experiments in mouse models have shown that dysregulation of one or a few critical genes in neurons can be sufficient to induce dysfunction and neurological symptoms, and the current evidence does not seem sufficient to rule it out. Figure 3D also suggests that PTGs in PV neurons may be particularly important, even accounting for the additional regions present in the RP analysis.

We agree with the reviewer that few critical neuronal genes might be sufficient to induce neurological dysfunction and symptoms and have added this point to the Results and Discussion. Additionally, we have highlighted that many neuronal genes are glia-enriched and might reflect glia contamination.

Key Conclusion 2: "Cell-type specific changes become only evident at late PrD stages." This conclusion is well supported.However, as the authors noted, due to legal constraints their model represents early to mid-disease onset rather than a true terminal environment matching that of patients. Therefore, it would be advantageous to choose a more appropriate name for the "terminal" group, perhaps based on one of the key humane endpoint criteria that would help readers in the field to place these important results in context of the overall disease process.

We have added additional information to clarify our definition of terminal stage to the Materials and methods.

Key Conclusion 3: "This suggests that the prion-induced molecular phenotypes reflect major glia alterations, whereas the neuronal changes responsible for the behavioral phenotypes may be ascribed to biochemically undetectable changes such as altered neuronal connectivity." The authors should modify the second half of this claim.As discussed above, changes to even a few neuronal genes can be sufficient to induce neurodegeneration. The claim that "the neuronal changes responsible for the behavioral phenotypes may be ascribed to biochemically undetectable changes," fails to acknowledge the changes in PV neurons observed in this study, however few they may be. The authors also do not take into account the possible role of transcribed RNAs that are not immediately translated (for example those that accumulate at synapses for fast translation on demand) or the overall proteome, which are not included in their analysis. Though their method cannot detect these components, the authors should examine the implications that such other changes may still be present in the discussion.The authors should also discuss the functions of the few specific PV PTGs and explore their potential relationship with neurodegeneration. This is especially important since the authors acknowledge that a key reason for including PV neurons in the analysis is ample evidence in the literature that they play a role in disease pathogenesis.Finally, the authors note that a top GO term in microglial cells was synaptic transmission. The authors should expand on this finding in the discussion, as the interplay of glia and neurons in the pathogenesis of disease is likely highly relevant.

We have removed the claim that “behavioral phenotypes may be ascribed to biochemically undetectable changes” and added the point that few neuronal changes might be sufficient to induce neuronal dysfunction and symptoms.

As stated in the manuscript, we believe that the enrichment of the GO term synaptic transmission in microglia is an artefact. We therefore refrained from further discussing this finding and have highlighted that it is in artefact in the results.

The data and methods are largely reproducible. Additional information should be provided about the methods for Gene Ontology analysis, how it was controlled, and what was used as a significance measure.

We have added additional information about the GO analysis to the Materials and methods section. The complete list of GO terms is now included as Supplementary file 10.

Some groups contain only two animals. At least three should be included per group for a minimally robust analysis.

We have tried to include 3 replicates per group as suggested by the reviewer. In few exceptions, we lost an individual sample and one sample had to be excluded due to low quality. In these instances (GFAP_2wpi Ctrl; CamKIIa_CX_term_Ctrl, CamKIIa_CX_term_PrD, Cx3cr1_term_Ctrl and Cx3cr1_term_PrD) we ensured that both replicates showed a high correlation and could still yield reliable results (see Author response image 1). Consistently, the DESeq2 algorithm (which can handle also just 2 replicates per group) identified differentially translated genes in the terminal samples.

**Author response image 1. sa2fig1:** Scatterplots of samples with just two replicates per group. Both ctrl (left panel) and the PrD (right panel) terminal CamKlla cortex samples showed a high correlation of rlog transformed RPF counts between replicates.

Reviewer #3 (Significance):This work provides an important conceptual advance in prion disease research that glia may be primary drivers of disease equal to or surpassing certain neuronal populations. Though the authors have shown previously that glial changes are dominant in bulk sequencing of the hippocampus, cell type-specific analysis adds an important level of detail to convince the field that few transcriptional changes occur in neurons though neurological defects are already present. Historically, neuronal defects have been assumed to occupy the main role, with glia being largely ignored. This echoes recent similar changes in other areas of the neurodegenerative disease field where we are recognizing the important roles of glia in pathogenesis, and how they may be modulated to treat disease.Their findings in PV neurons also may reflect early key changes in this important neuronal population that contribute to neurological symptom onset. They will allow further study of the genes and pathways involved and may lead to additional effective treatments for disease.Finally, the thorough comparison of multiple neuronal and glial populations will allow future investigation of the interplay of neurons and microglia in pathogenesis and shows the importance of studying them synergistically rather than individually.Audience:The neurodegenerative disease field in general will be interested in the findings. Immunologists, other neuroscientists, and pharmaceutical and other drug development organizations will also be influenced by the work.Reviewers cross-commenting:I agree with reviewer 1 that a comparison of the total transcriptome with ribosomally active transcripts would aid the interpretation of this work. It would also uncover or refute the presence of cell-type differences in translation efficiency that directly impact the authors' major conclusion that glia are more affected than neurons. I support the request of this additional experiment.

As discussed above we have refrained from such a comparison since 1) the scope of this study was to identify biologically relevant prion-induced molecular changes and not study post-transcriptional regulation, 2) the generation of such dataset will take ~ 2 years, and 3) difference between transcriptional and translational changes are likely a combination of post-transcriptional regulation and artefact induced change that are probably difficult to interpret.